# A combination strategy targeting enhancer plasticity exerts synergistic lethality against BETi-resistant leukemia cells

Lei Guo[1,11], Jia Li[1,11], Hongxiang Zeng[1], Anna G. Guzman[2,3], Tingting Li[1], Minjung Lee[1], Yubin Zhou [4],
Margaret A. Goodell[2,3,5,6], Clifford Stephan[4], Peter J.A. Davies[4], Mark A Dawson[7,8,9,10], Deqiang Sun[1,12]* &
Yun Huang [1,12]*

Primary and acquired drug resistance imposes a major threat to achieving optimized clinical outcomes during cancer treatment. Aberrant changes in epigenetic modifications are closely involved in drug resistance of tumor cells. Using BET inhibitor (BETi) resistant leukemia cells as a model system, we demonstrated herein that genome-wide enhancer remodeling played a pivotal role in driving therapeutic resistance via compensational re-expression of pro-survival genes. Capitalizing on the CRISPR interference technology, we identified the second intron of lncRNA, *PVT1*, as a unique bona fide gained enhancer that restored *MYC* transcription independent of BRD4 recruitment in leukemia. A combined BETi and CDK7 inhibitor treatment abolished *MYC* transcription by impeding RNAPII loading without affecting *PVT1*-mediated chromatin looping at the *MYC* locus in BETi-resistant leukemia cells. Together, our findings have established the feasibility of targeting enhancer plasticity to overcome drug resistance associated with epigenetic therapies.

[1] Center for Epigenetics & Disease Prevention, Institute of Biosciences and Technology, Texas A&M University, Houston, TX 77030, USA. [2] Stem Cells and Regenerative Medicine Center, Baylor College of Medicine, Houston, TX 77030, USA. [3] Center for Cell and Gene Therapy, Baylor College of Medicine, Houston, TX 77030, USA. [4] Center for Translational Cancer Research, Institute of Biosciences and Technology, Texas A&M University, Houston, TX 77030, USA. [5] Department of Pediatrics, Section of Infectious Diseases, Baylor College of Medicine, Houston, TX 77030, USA. [6] Program in Developmental Biology, Baylor College of Medicine, Houston, TX 77030, USA. [7] Peter MacCallum Cancer Centre, Melbourne, VIC, Australia. [8] Sir Peter MacCallum Department of Oncology, University of Melbourne, Melbourne, VIC, Australia. [9] Department of Hematology, Royal Melbourne Hospital & Peter MacCallum Cancer Centre, Melbourne, VIC, Australia. [10] Centre for Cancer Research, University of Melbourne, Melbourne, VIC, Australia. [11] These authors contributed equally: Lei Guo, Jia Li. [12] These authors jointly supervised this work: Deqiang Sun, Yun Huang *email: dsun@tamu.edu; yun.huang@tamu.edu

D rug resistance can arise from transcription reactivation, bypass and alterations during anticancer treatment[1–3]. Transcriptional adaptation during drug treatment is often mediated by inducible histone modifications, especially histone H3 lysine 27 acetylation (H3K27ac) at the distal enhancer elements, thus activating the transcription of drug resistance-associated genes[4–6]. BRD4 (bromodomain-containing protein 4), a member of the bromodomain and extra-terminal domain (BET) family, acts as a chromatin reader to regulate transcription by linking histone acetylation and core components of the transcriptional apparatus[7]. BET inhibitors (BETi), as exemplified by JQ1 and I-BET151, have been shown to suppress the growth of multiple types of tumor both in vitro and in vivo[8]. However, potential drug resistance associated with BETi becomes one of the major hurdles hampering the clinical applications of these promising drug candidates[8,9].

As one of the most critical cis-regulatory elements in the mammalian genome, enhancers provide binding sites for transcription factors (TFs) and often interact with promoters of target genes over long genomic distances—ranging from several to hundreds of kilobases—to enable tissue-specific transcription[10,11]. For example, the blood enhancer cluster (BENC) super enhancer has been shown to modulate the expression of a proto-oncogene MYC in cells of the hematopoietic system and account for the maintenance of MLL-AF9 driven leukemia[12,13]. In addition, recent studies have unveiled that, although the promoter of Plasmacytoma variant translocation 1 (PVT1) is a tumor-suppressive chromatin boundary element, several intragenic regions in the PVT1 locus act as enhancers to regulate MYC expression via chromatin looping in a tissue-specific manner[14,15]. However, the function of PVT1 intragenic enhancers in BETi-resistant leukemia cells remains largely unexplored.

While enhancer-promoter looping is important for active transcription, RNA polymerase II (RNAPII) is the enzyme directly involved in the control of transcriptional activity in human and rodents. A high correlation of RNAPII occupancy and the chromatin domain architectures has been documented in Drosophila, suggesting that transcriptional status might define chromatin domains and compartments[16]. During oncogenesis, tumor cells undergo transcriptional reprogramming by remodeling enhancer-promoter looping[17,18]. Therefore, RNAPII loading and elongation at remodeled enhancer-promoter looping regions might add an additional layer of control over aberrant transcription of oncogenes in drug resistant cells[19]. Targeting the transcriptional activity on remodeled enhancer regions might be explored as an alternative strategy to overcome therapeutic resistance.

In this study, we discovered a strong synergistic lethality of targeting the reprogrammed enhancers in BETi-resistant leukemia cells by suppressing BRD4 and CDK7 activity in vitro and in vivo. Using both human and mouse BETi-sensitive or BETi-resistant leukemia cells, we identified a BETi-resistance specific enhancer within the PVT1 locus, which facilitated MYC expression in BETi-resistant cells. This BRD4-independent de novo enhancer restored the enhancer-promoter looping at the MYC locus to drive MYC transcription in BETi-resistant leukemia cells. Suppressing the RNAPII activity by cyclin-dependent kinase 7 (CDK7) inhibitor interrupted RNAPII loading at this BRD4-independent de novo enhancer-promoter looping region, thereby suppressing the growth of BETi-resistant malignant cells. Overall, our study has established the preclinical rationale for targeting enhancer plasticity to overcome BETi resistance in cancer cells.

values of JQ1, a well-known bromodomain inhibitor with high potency against BRD4, in a panel of cancer cell lines derived from leukemia ($n = 26$), including chronic myeloid leukemia (CML), T-lineage acute lymphoblastic leukemia (T-ALL), acute myeloid leukemia (AML), and erythroleukemia, based on the publicly available database PHARMACODB[20] (Fig. 1a). AML cell lines were highly sensitive to JQ1, whereas T-ALL (e.g., Jurkat) and CML cell lines (e.g., K562 and MOLT-3) were innately resistant to BETi with their $IC_{50}$ values in the micromolar ranges. To explore the enhancer landscapes of leukemia cells exhibiting contrasting sensitivities to BETi, we analyzed the ChIP-seq data of H3K27ac, an enhancer mark, in BETi-sensitive (MV4-11, THP1, MOLM-13) and BETi-resistant (K562, MOLT-3, Jurkat) cells, respectively (Supplementary Data 1, Supplementary Fig. 1a). Distinct H3K27ac patterns were observed in BETi-resistant cells but not in BETi-sensitive cells, as reflected in the principal component analysis (PCA) plot (Fig. 1b). This finding indicated that intrinsic enhancer landscapes might contribute to primary BETi resistance.

In parallel, we took advantage of a BETi-resistant murine AF9 AML cell line, in which drug resistance was gradually established by long-term culture with escalating BETi doses, as our acquired resistance model[21]. A matched parental non-resistant cell line was used as a BETi-sensitive control. We found that H3K27ac-enriched regions were remodeled dramatically in the resistant cell line when compared to the parental non-resistant cells. In fact, 66,952 genomic regions were identified with differential enrichment of H3K27ac between the parental and resistant murine AF9 AML cells (Fig. 1c, Supplementary Data 1). Since BRD4 binds to acetylated histones as epigenetic reader and plays critical roles at enhancer regions, we further examined the BRD4 distribution in these differentially enriched H3K27ac regions. Surprisingly, 21,700 (33.9%) differentially enriched regions displayed increased H3K27ac deposition but showed no change or decreased Brd4 binding in BETi-resistant cells compared with parental cells (Fig. 1d), suggesting that genomic regions displaying increased H3K27ac levels under a BETi-resistant state were less dependent on BRD4 binding. More interestingly, these H3K27ac-enriched but Brd4-independent regions in BETi-resistant cells were mainly located at distal-regulatory enhancer regions with an average size of 30 kb to the nearest promoters (Fig. 1e). We subsequently performed Genomic Regions Enrichment of Annotations Tool (GREAT) analysis to reveal the implicated functions of these distal H3K27ac-enriched but Brd4-independent regions[22]. The majority of them were closely associated with genes that are important for myeloid cell differentiation and function (Fig. 1f). If not taking into account Brd4 binding, genomic regions displaying H3K27ac enrichment were associated with genes that are involved in the regulation of not only myeloid cell function but also hematopoiesis and lymphocyte differentiation (Supplementary Fig. 1b).

To further examine whether enhancer remodeling is a common feature in BETi-resistant tumors other than hematological malignancies, we analyzed the publicly available H3K27ac ChIP-seq data within matched neuroblastoma and breast cancer cells[23,24]. Indeed, we observed dynamic changes of H3K27ac enrichment between parental and BETi-resistant cells (Supplementary Fig. 1c–e). We further compared genomic regions showing notable changes of H3K27ac and BRD4 enrichment between BETi-resistant cells and the matched parental non-resistant tumor cells. Over 60% of genomic regions displayed increased H3K27ac enrichment but showed no change or reduction of BRD4 occupancy in the BETi-resistant group (Supplementary Fig. 1d, e, right panel).

## Results

**BRD4-independent enhancer remodeling in BETi-resistant cells.** In order to study BETi resistance, we first ranked the $IC_{50}$

**Synthetic lethality of BET and CDK7 dual inhibition.** The above findings strongly suggested that BRD4-independent

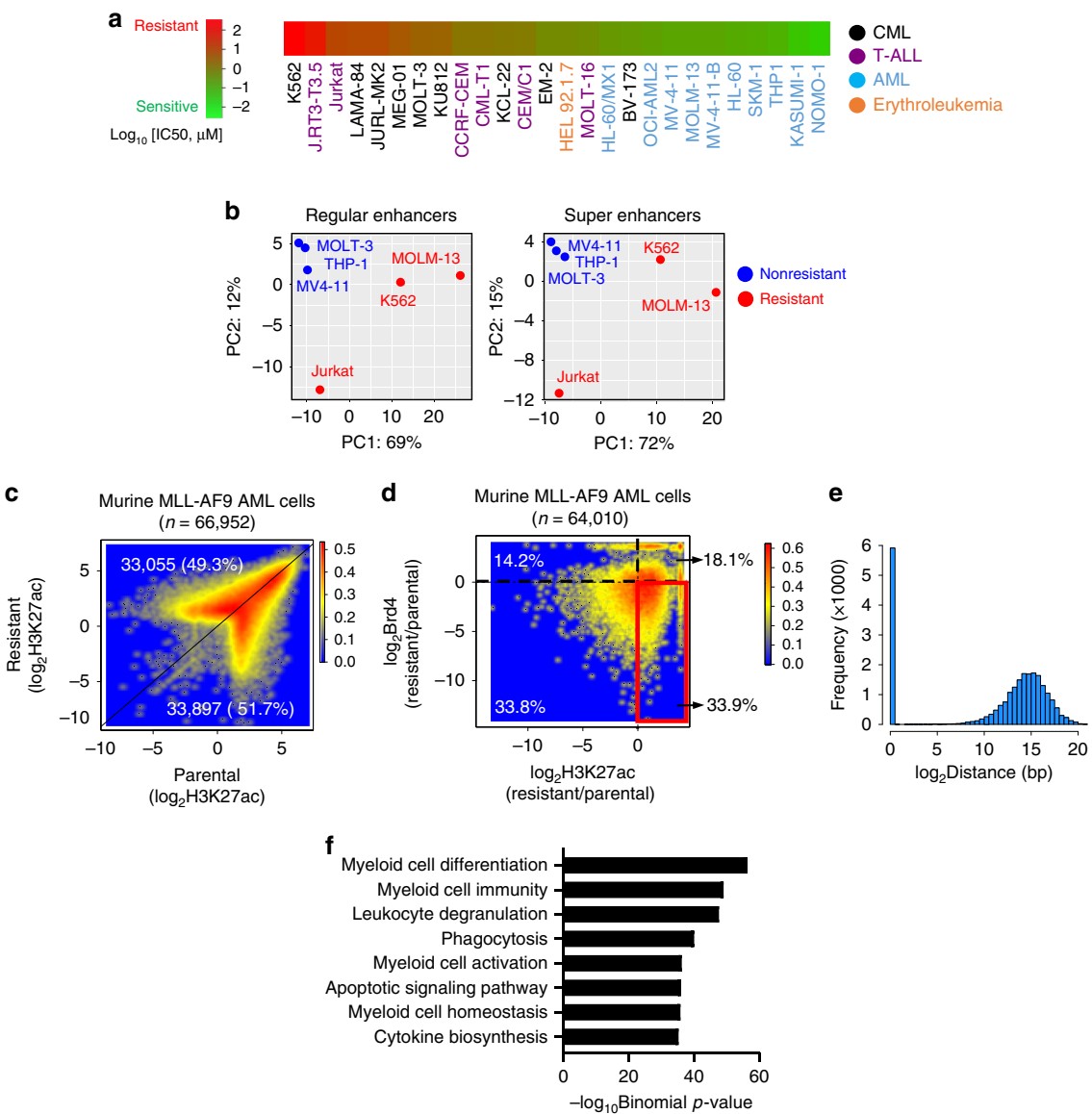

**Fig. 1 BETi-resistant leukemia cells undergo BRD4-independent enhancer remodeling. a** Heatmap representing the $IC_{50}$ values of the indicated leukemia cell lines to a BRD4 inhibitor JQ1. The $IC_{50}$ values were obtained from the published database PHARMACODB. **b** The principal component analysis (PCA) of enhancer region distributions in the indicated BETi-sensitive and BETi-resistant leukemia cells. **c** Smoothed scatter plot representation of H3K27ac-enriched regions in paired BETi-sensitive (parental, x-axis) and BETi-resistant (y-axis) murine AF9 AML cells. Color code represents the dot density. **d** Smoothed scatter plot representation of differential H3K27ac-enriched regions and Brd4 enriched regions in paired BETi-sensitive (parental) and BETi-resistant murine AF9 AML cells. Red highlighted region: genomic regions displayed increased H3K27ac enrichment with no change or decreased Brd4 binding in BETi-resistant murine AF9 AML cells. All the publicly available H3K27ac and BRD4 ChIP-seq datasets used in Fig. 1b–d were summarized in Supplementary Table 1. Color code represents the dot density. **e** The histogram showing the distribution of distance between the genomic regions highlighted within the red box in Fig. 1d and their closest genes. **f** Genomic Regions Enrichment of Annotations Tool (GREAT) analysis of genomic regions highlighted in the red box of Fig. 1d.

enhancer remodeling might be one of the common mechanisms exploited by multiple BETi-resistant tumor cells to drive transcriptional reprogramming. This prompted us to explore anticancer strategies by targeting BETi-resistant associated enhancers. We carried out a limited combination therapy screen with two BETi-resistant leukemia cell lines (K562 and Jurkat) by combining BETi with selected chemical inhibitors that are known to modulate epigenetic modifications and enhancer activity (Supplementary Fig. 2)[25–29]. THZ1 as a CDK7 inhibitor stood out as the top hits and showed enhanced cytotoxicity when combined with I-BET151 (a BRD4 inhibitor) (Supplementary Fig. 2).

Given that CDK7 inhibitors have been shown to suppress pro-oncogenic transcription by targeting enhancers in tumor cells[30–35], we envisioned that dual inhibition of BRD4 and CDK7 might synergize to suppress the growth of BETi-resistant leukemia cells to overcome resistance. To validate this possibility, we carried out a well-established combinatorial assay on leukemia cells with increasing doses of I-BET151 (a BRD4 inhibitor) and/or THZ1 (a CDK7 inhibitor). Upon dual inhibition of BRD4 and CDK7 using the combination of I-BET151 and THZ1, we observed a strong synergistic lethality in K562 and Jurkat leukemia cells (Fig. 2a, b), but not in OCI-AML2 cells that are highly sensitive to

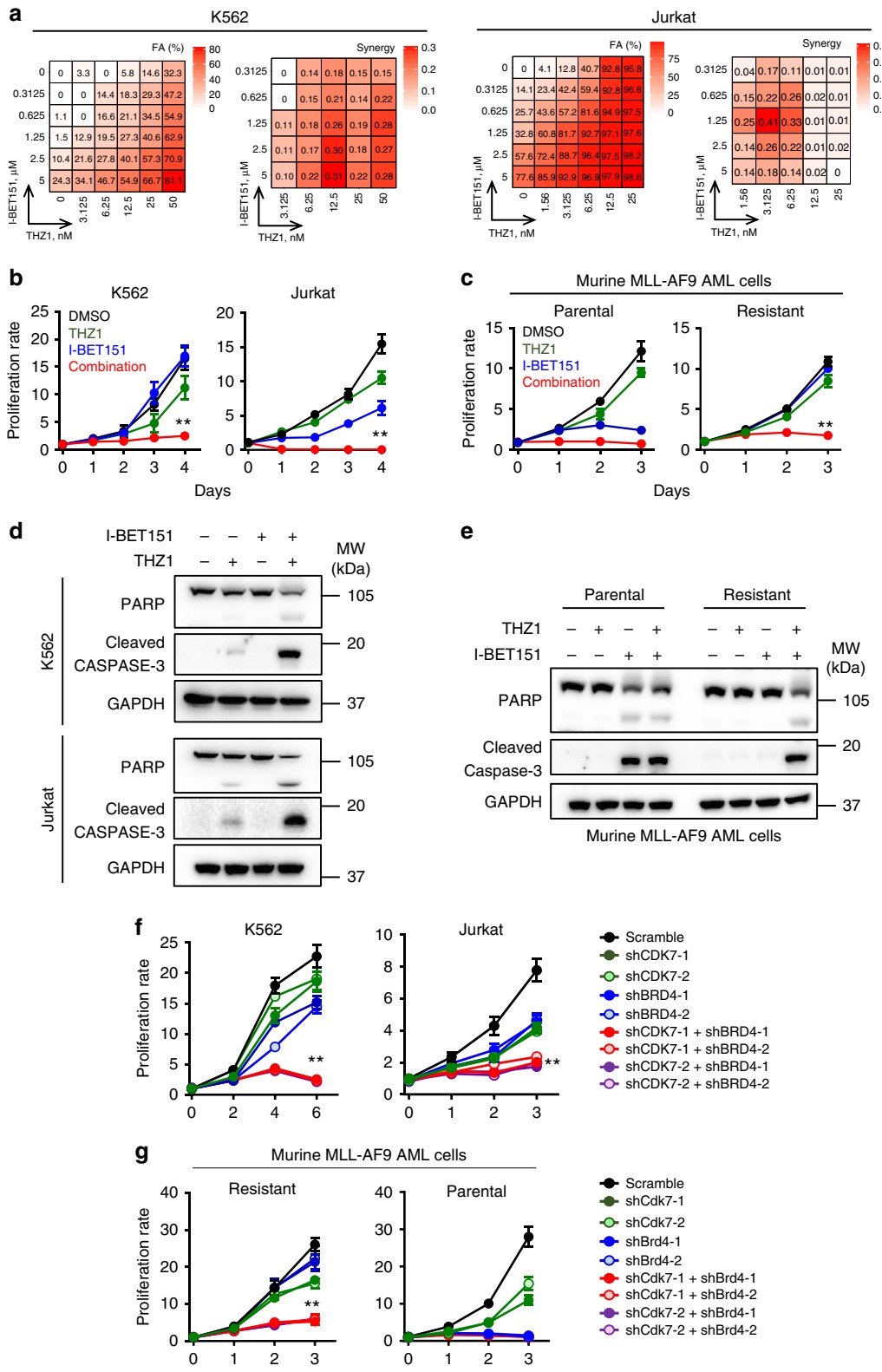

BETi ($IC_{50} = 160$ nM)[20] (Supplementary Fig. 3a). The synthetic lethality imposed by I-BET151 and THZ1 was also observed in murine AF9 AML cells that were resistant to BETi, but not in the matched non-resistant parental cells (Fig. 2c). Further flow cytometry and immunoblot analyses revealed that dual inhibition of BET and CDK7 caused cell cycle arrest at the G1 phase (Supplementary Fig. 3c) and enhanced apoptosis in both human leukemia cell lines (K562 and Jurkat, Fig. 2d and Supplementary Fig. 3d) and murine BETi-resistant AF9 AML cells (Fig. 2e and Supplementary Fig. 3e). Furthermore, we observed similar synthetic lethality by using other BET inhibitors, including JQ1, OTX-015, and I-BET762, in combination with THZ1 in BETi-resistant human and murine leukemia cells (Supplementary Fig. 4).

**Fig. 2 Dual inhibition of CDK7 and BRD4 synergistically suppresses the growth of BETi-resistant leukemia cells. a** Heatmap representation of Fraction Affected (FA) and Bliss interaction index across five-point dose range of a BET inhibitor (I-BET151) and a CDK7 inhibitor (THZ1) in K562 and Jurkat cells. Mean values of triple biological experiments were shown. (b and c) Proliferation analysis of K562 and Jurkat cells (**b**) or murine AF9 AML cells (**c**) treated with DMSO (black), THZ1 (green), I-BET151 (blue), and the combination of THZ1 + I-BET151 (red). The concentrations of I-BET151 and THZ1 were 2.5 μM and 12.5 nM for K562, 2.5 μM and 3.125 nM for Jurkat, 2.5 μM and 50 nM for murine AF9 AML cells, respectively. Data were shown as mean ± S.D; $n = 6$ from 3 independent assays, **$P = 0.00002$ (K562), 0.00009 (Jurkat) and 0.000006 (AF9 resistant), by two-tailed Student's $t$ test. **d, e** Immunoblot analysis on apoptosis-related marker PARP and cleaved caspase 3 (C/Caspase3) in K562 (**d**, top), Jurkat (**d**, bottom) and murine AF9 AML cells (**e**) treated with DMSO, THZ1, I-BET151, and the combination of THZ1 + I-BET151. The inhibitor concentrations were the same as shown in Fig. 2b, c. Three independent assays were performed. **f, g** Quantification of proliferation of K562, Jurkat cells (**f**) and murine AF9 AML cells (**g**) transduced with shRNAs targeting CDK7 and/or BRD4. Data were shown as mean ± S.D; $n = 8$ from three independent assays, **$P = 0.00003$ (K562), 0.00001 (Jurkat) and 0.000002 (AF9 resistant), by two-tailed Student's $t$ test.

To rule out the possibility that the observed inhibitory effect might arise from off-target effects of chemicals, we knocked down BRD4 and CDK7 individually or in combination with shRNAs in both human (K562 and Jurkat) and murine AF9 leukemia cells (Supplementary Fig. 5). Consistent with the results from pharmacological inhibition using BETi and/or THZ1, we found that only the dual knockdown of BRD4 and CDK7, but not single-knockdown, substantially inhibited the growth of BETi-resistant leukemia cells (red/purple curves; Fig. 2f, g). By contrast, single knockdown of BRD4 was sufficient to suppress the growth of BETi-sensitive leukemia cells in vitro (blue curves; Fig. 2f, g). Together, results from both pharmacological inhibition and genetic depletion studies converge to support the conclusion that co-inhibition of BET and CDK7 imposes synergistic lethality against both human and rodent BETi-resistant leukemia cells in vitro.

To further validate the synthetic lethality in vivo, we adoptively transferred BETi-resistant murine AF9 AML cells into sub-lethally irradiated CD45.1 recipient mice, followed by treatment with I-BET151 and THZ1, individually or in combination, for up to 5 weeks (Fig. 3a). Consistent with the in vitro data, only recipient mice receiving the combination treatment showed the most effective therapeutic outcomes, as characterized by prolonged overall survival (Fig. 3b) and reduced tumor burdens in the spleen and bone marrow (Fig. 3c–f). Compared with the control (DMSO) or single-treatment (I-BET151 or THZ1 alone) groups, the combination treatment group showed less severe splenomegaly (Fig. 3c) without significant changes of the overall body weight (Supplementary Fig. 6a), accompanied with a pronounced reduction of transferred AF9 AML cells (YFP-positive) in both the spleen and bone marrow after 2-week treatment (Fig. 3d, Supplementary Fig. 6b). In line with reduced tumor burdens in the spleen and bone marrow, recipient mice treated with BETi and THZ1 had less AML cells in the peripheral blood (Fig. 3e), along with attenuated infiltration of tumor cells in the liver (Fig. 3f). In these recipient mice, the morphology of spleen and bone marrow after adoptive transfer remained relatively normal (Fig. 3f). To further evaluate the potential toxicity associated with the BETi + THZ1 combination, we performed histological analyses on major organs from normal mice treated with DMSO (control) or the combination therapy (BETi + THZ1). Both groups did not exhibit overt histological abnormality in tissues collected from heart, lung, spleen, kidney, and small intestine (Supplementary Fig. 6c). However, mild liver damage, characterized by histopathological changes in parenchyma, was noted in mice after the combination therapy for 20 days (Supplementary Fig. 6d). The liver function measured by serum alanine aminotransferase (ALT) levels further confirmed liver damage in mice treated with the combination therapy (Supplementary Fig. 6e).

Next, we evaluated the toxicity of the combination therapy in the hematopoietic system. We observed a mild decrease of HSPCs

(lin-Sca1 + cKit+, LSK) and HSCs (lin-Sca1 + cKit + CD150 + CD48-) in the bone marrows of mice receiving the combination therapy for 2 and 5 weeks, as well as in the in vitro colony-forming unit (CFU) assay (Supplementary Fig. 6f–i). Using a complete blood count (CBC) analysis, no obvious adverse effect was observed in the hematopoietic system in the mice treated with the combination therapy for 2 weeks. However, in the moribund mice after 5 weeks treatment, we observed reduction of white blood cell, red blood cells, platelets in peripheral blood and decreased T cell, B cell and myeloid cells in bone marrow with a trace of YFP + tumor cells (Supplementary Fig. 6b–i and Supplementary Data 2), suggesting the potential toxicity of the combination therapy to the hematopoietic system for the long-term treatment. Nevertheless, the growth suppressive effects imposed by the BETi and THZ1 combination treatment was much more pronounced in BETi-resistant AML cells when compared with their effects on normal blood cells (Supplementary Fig. 6b–h).

Collectively, we have demonstrated that co-inhibition of BET and CDK7 exerts synthetic lethality over BETi-resistant leukemia in vivo. Within a short-term treatment (within 2 weeks) window, the toxicity to liver and the hematopoietic system was not obvious. However, for long-term treatment (after 5 weeks), the combination therapy led to undesired liver toxicity and induced suppressive effects on the hematopoietic system. Hence, further optimization of the leading drug candidates is required to reduce the off-tumor toxicity associated with BETi and THZ1 inhibitors for future clinical applications.

**BET and CDK7 synergistically suppresses MYC-associated genes.** We next moved on to dissect the molecular mechanisms underlying the observed synthetic lethality in BETi-resistant leukemia cells. We performed transcriptomic analyses using RNA-seq on K562, Jurkat and Murine AF9 AML cells treated with DMSO, I-BET151, THZ1 and the combination of THZ1 + I-BET151. The transcriptomic signatures of these four groups were far separated in PCA biplots, indicating distinct gene expression profiles upon treatment. However, the relative positions of each group remained similar in all three biplots, suggesting the similarity of transcriptional patterns for each treatment among the three examined leukemia cell types (Fig. 4a). After comparing the differentially expressed genes (DEGs) in the four groups (DMSO, I-BET151, THZ1 and combination) in K562 cells, we identified 2382 down-regulated and 1877 up-regulated DEGs that were synergistically perturbed in the combination group (Fig. 4b). Similarly, 825 down-regulated and 782 up-regulated DEGs were uniquely detected in the murine AF9 AML cells (Fig. 4b). Gene set enrichment analysis (GSEA) further revealed that these affected genes were downstream of key TFs that are associated with oncogenic super enhancers, including MYC and E2F (Fig. 4c, d). With real-time quantitative PCR (qPCR), we further

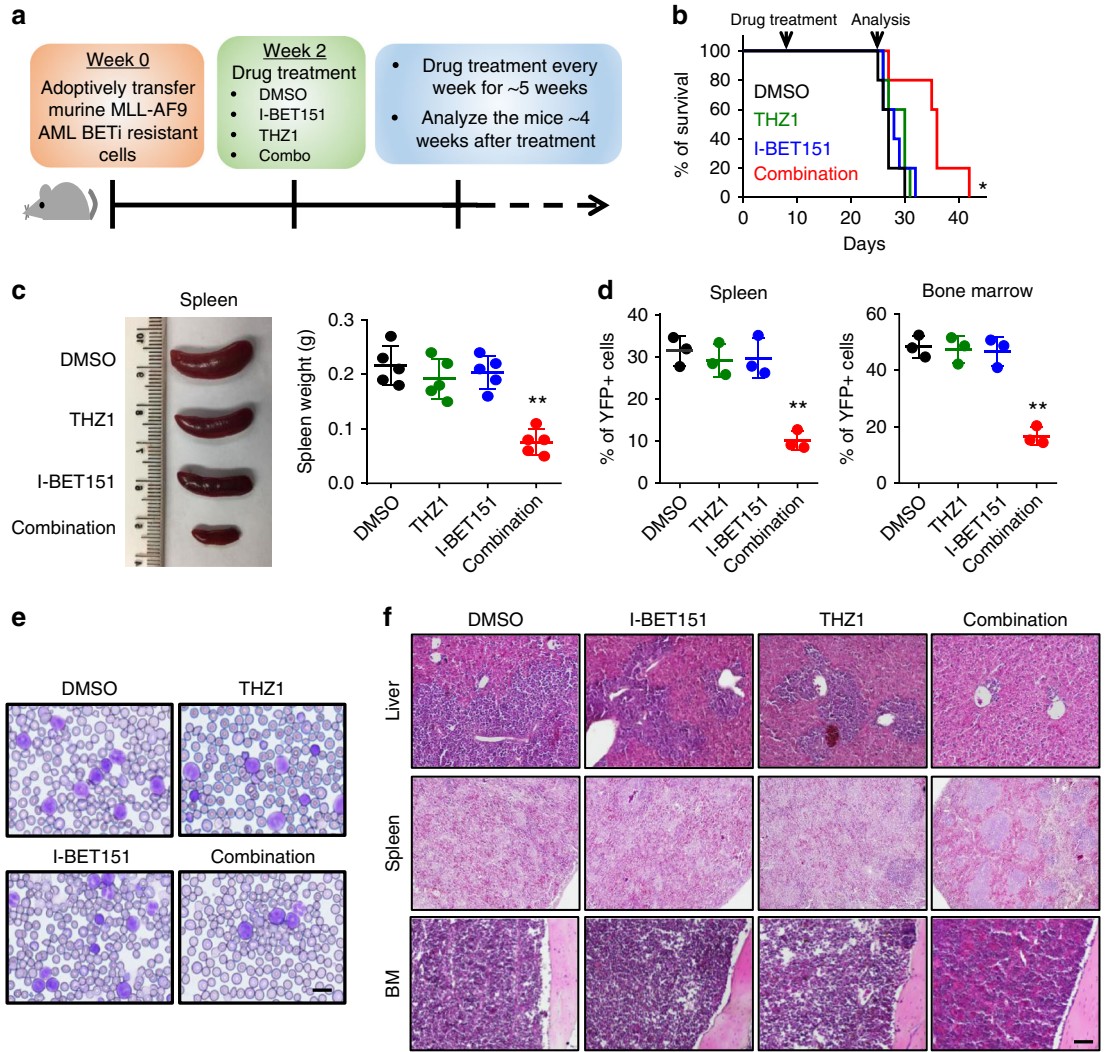

**Fig. 3 Synthetic lethality imposed by dual inhibition of CDK7 and BRD4 toward BETi-resistant leukemia cells in vivo. a** Schematic of the experimental design for CDK7 and / or BRD4 inhibitor treatment in vivo. **b** Kaplan–Meier survival curves of recipient mice transferred with BETi-resistant murine AF9 AML cells and then treated with DMSO (black), THZ1 (green, 10 mg/kg), I-BET151 (blue, 10 mg/kg), or a combination of THZ1 with I-BET151 (red). $n = 5$ mice. *$P = 0.0112$, by log rank Mantel-Cox test. **c** Representative images and weights of spleens collected from recipient mice transferred with BETi-resistant murine AF9 AML cells at 20 days after DMSO, THZ1, I-BET151, or combination treatment. $n = 5$ mice, $P = 0.0049$, by ANOVA with Dunnett's post-hoc correction. **d** Percentage of YFP+ murine BETi-resistant AF9 AML cells collected from the spleen and bone marrow of the mice treated with DMSO (black), THZ1 (green), I-BET151 (blue), or a combination of THZ1 with I-BET151 (red). Data were shown as mean ± S.D; $n = 3$ mice, **$P = 0.0044$, by two-tailed Student's $t$ test. **e** Representative Giemsa staining of the peripheral blood collected from recipient mice transferred with BETi-resistant murine AF9 AML cells at 20 days after DMSO, THZ1, I-BET151, or combination treatment of THZ1 and I-BET151. $n = 5$ mice. Scale bar, 10 μm. **f** Representative Hematoxylin and eosin (HE) staining of liver, spleen, and bone marrow tissues collected from the recipient mice transferred with BETi-resistant murine AF9 AML cells at 20 days after DMSO, THZ1, I-BET151, or the combination treatment. $n = 5$ mice. Scale bar, 100 μm.

examined the mRNA levels of *MYC*, *MYB*, *TAL*1 and *LMO2*, four key genes that are known to be regulated by super enhancers during leukemogenesis and disease progression[36–38]. We detected a marked reduction in the expression of all four super-enhancer regulated genes in K562 and murine BETi-resistant AF9 AML cells upon the combination treatment (Fig. 4e).

Since *MYC* and *E2F* targets ranked as the top hits in our GSEA analysis, we focused on *MYC* and *E2F* as examples to confirm the downregulation at the protein level with immunoblotting (Fig. 4f, Supplementary Fig. 7a). The dual suppression of BET and CDK7 significantly reduced the MYC protein levels in both BETi-resistant K562 and murine AF9 AML cells (Fig. 4f), but had no appreciable effects on the E2F protein levels (Supplementary Fig. 7a). Therefore, we focused on *MYC*

and its genomic targets for the following studies. Furthermore, consistent with earlier reports[30], we confirmed that inhibiting CDK7 could suppress the phosphorylation of RNAPII (serine 2, 5, 7 phosphorylation), without altering the total RNAPII and CDK7 protein levels (Fig. 4f). We also observed that inhibiting BET and CDK7 individually or in combination had no effects on the protein level of BRD4 (Fig. 4f). Similar expression patterns were observed in both K562 and murine AF9 AML cells after the combination treatment by using THZ1 with other BETi, including JQ1, OTX15 or I-BET762 (Supplementary Fig. 7b, c). To further confirm that MYC is a major target involved in suppressing the growth of BETi-resistant cells following the combination treatment, we performed rescue experiment by overexpressing MYC in K562 and murine AF9

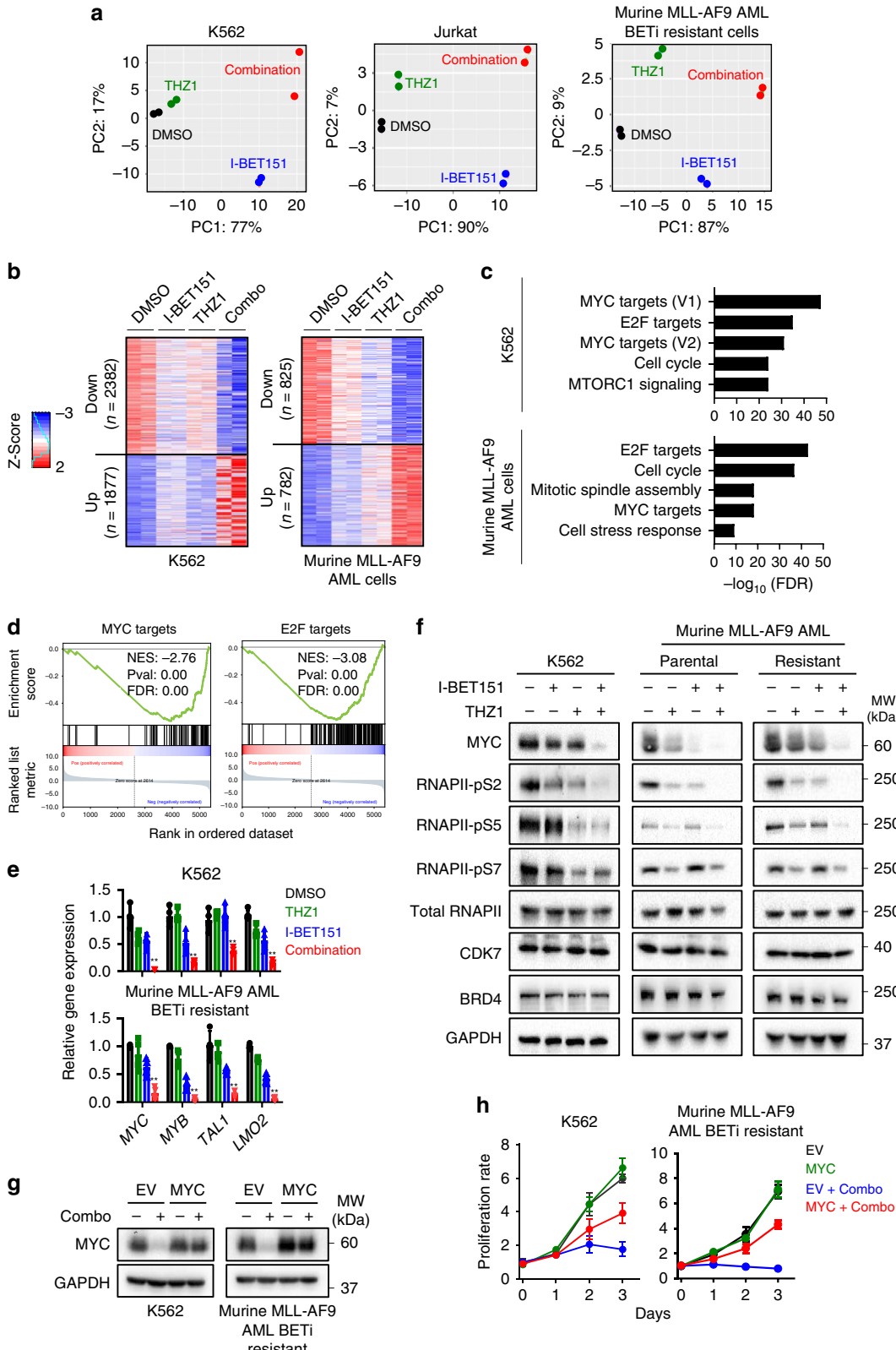

AML cells (Fig. 4g). MYC overexpression partially restored the cell growth (Fig. 4h) and overcame the cell cycle arrest at G1 (Supplementary Fig. 7d) in both types of leukemia cells treated with BETi + THZ1. Taken together, these data indicated that BETi and THZ1 synergistically inhibited the growth of BETi-resistant leukemia cells by suppressing the expression of super-enhancer regulated oncogenic genes, as notably exemplified by MYC.

**Enhancer remodeling restores MYC expression.** Because BETi led to strong suppression of MYC expression in BETi-sensitive

**Fig. 4 MYC identified as a critical target that mediates the synergistic lethality of I-BET151 and THZ1 in BETi-resistant leukemia cells. a** The PCA analysis of RNA-seq results in K562, Jurkat and BETi-resistant murine AF9 AML cells treated with DMSO (black), I-BET151 (blue), THZ1 (green), and the combination of I-BET151 + THZ1 (red) for 24 h. **b** Heatmap representation of differentially expressed genes (DEGs) identified between the DMSO, I-BET151, THZ1, and the combination treatment in K562 and murine AF9 BETi-resistant AML cells. DEGs were defined as q-value <= 0.05. Red and blue color stand for up- and down-regulated genes, respectively. (**c**) Gene Set Enrichment Analysis (GSEA) of DEGs identified from Fig. 4b. **d** GSEA presentation of MYC or E2F targeted genes in the identified DEGs. Genes were ranked by fold changes. **e** Real-time qPCR analysis of super-enhancer related genes in K562 and murine BETi-resistant AF9 AML cells after DMSO (black), I-BET151 (blue), THZ1 (green), and the combination treatment (red) for 24 h. Data were shown as mean ± S.D; n = 4 from four independent assays. **P = 0.002 (MYC), 0.0015 (MYB), 0.01 (MEIS1) and 0.0006 (LMO2), by two-tailed Student's t test. **f** Representative Western blotting showing the protein levels of MYC, CDK7, BRD4, phosphorylated and total RNAPII in K562, murine BETi-sensitive or resistant AF9 AML cells after DMSO, I-BET151, THZ1, and the combination treatment for 24 h. GAPDH was used as control. n = 3 from 3 independent assays. **g** Immunoblot analysis of MYC expression in K562 and BETi-resistant murine AF9 AML cells transduced with the empty vector (EV) or a lentivirus encoding MYC. Cells were treated with DMSO or combination (Combo) for 24 h. Three independent assays were performed. **h** Proliferation rate of K562 and BETi-resistant murine AF9 AML cells expressing the empty vector (EV) or MYC after DMSO or the combination treatment (Combo) for 3 days. Data were shown as mean ± S.D; n = 8 from 4 independent assays. Drug concentrations used in Fig. 4 were the same as in Fig. 2b, c for each cell line.

but not in BETi-resistant cells (Fig. 4f and Supplementary Fig. 7c), we reasoned that a compensatory mechanism associated with enhancer remodeling might restore MYC expression in BETi-resistant cells. To identify the enhancers that would enable MYC re-expression in BETi-resistant leukemia cells, we examined the H3K27ac peak intensities in MYC-associated super enhancers[13] among leukemia cell lines with high and low IC50 values to BETi (Fig. 5). We discovered that the second intron of the PVT1 locus displayed the most significant H3K27ac enrichment in BETi-resistant cell lines (K562, Jurkat, MOLT-3; red traces) compared with BETi-sensitive cells (MV4-11, MOLM-13, and THP1; blue traces) (Fig. 5a, b). Worthy to note, the H3K27ac peaks within the well-established BENC super enhancer (E1-E5 associated with MYC expression) were all attenuated in leukemia cell lines with high IC50 values for BETi (red traces; Fig. 5b, right panel). In addition, we observed a gained H3K27ac peak at the same intron of the PVT1 locus in the murine BETi-resistant AF9 AML cells, but not in the parental non-resistant cells (Fig. 5c). H3K27ac enrichment at these regions were further independently confirmed by H3K27ac ChIP-qPCR in murine BETi-resistant AF9 AML cells (Supplementary Fig. 8a). Interestingly, we observed BRD4 enrichment at the BENC enhancer regions, but not at the PVT1 locus, in both parental and BETi-resistant AF9 AML cells, suggesting that PVT1 enhancer might be independent on BRD4 binding and take control over MYC re-expression after acquisition of BETi resistance (Fig. 5c). We next moved on to examine the chromatin accessibility of this H3K27ac peak within the PVT1 locus by comparing the Assay for Transposase-Accessible Chromatin using sequencing (ATAC-seq) data between K562 (high BETi IC50: 10 µM) and THP1 cells (low IC50: 60 nM). A strong ATAC-seq peak was detected in K562 cells, but not in THP1 cells (Fig. 5d), which overlapped with the gained H3K27ac peak shown in Fig. 5b. The PVT1 locus is known to contain multiple intragenic enhancers to regulate MYC expression by enhancer-promoter looping in a cell type-specific manner[14,15]. We therefore further analyzed the H3K27ac ChIP-seq data in paired parental and BETi-resistant neuroblastoma and breast cancer cells. However, in these two cancer types, we did not observe any de novo H3K27ac enrichment (Supplementary Fig. 8b), suggesting that the enriched H3K27ac peak at the second intron of PVT1 might be a gained BETi-resistant enhancer specific for leukemia.

To further determine whether this de novo H3K27ac peak within the PVT1 locus is essential for maintaining MYC expression in BETi-resistant leukemia cells, we resorted to the CRISPR interference (CRISPRi) technique by fusing a catalytically-inactive Cas9 (dCas9) with a transcriptional repressor KRAB (the Krüppel-associated box; dCas9-KRAB)[39]. dCas9-KRAB enabled us to site-specifically edit the H3K27ac

level of individual MYC enhancers, including PVT1 and BENC (Fig. 6a). KRAB can recruit the NuRD/HDAC complex to reduce the acetylation of H3K27, thereby suppressing gene transcription[40]. To confirm the targeting efficiency at the desired loci, we first performed H3K27ac ChIP-qPCR at MYC super enhancers in cells transduced with dCas9-KRAB and sgRNAs targeted to individual enhancers. Indeed, we observed strong reduction of H3K27ac enrichment at each of the sgRNA-targeted MYC enhancers in K562 cells expressing dCas9-KRAB/sgRNAs (Fig. 6b). Next, to examine whether reduced H3K27ac enrichment at these enhancers altered MYC transcription, we compared MYC expression in K526 cells in the absence or presence of I-BET151 following CRISPRi at enhancers. When dCas9-KRAB was targeted to the PVT1 locus in BETi-treated K562 cells, we noted a 67% reduction in the mRNA level of MYC (Fig. 6c), as well as a marked reduction of MYC protein expression (Fig. 6d). Functionally, downregulation of MYC expression was accompanied with reduced viability of BETi-treated K562 cells (Fig. 6e). As stringent control, I-BET151-treated K562 cells upon CRISPRi with scrambled sgRNAs did not induce any significant changes in MYC expression and cell viability (Fig. 6c–e). In addition, regardless of I-BET151 treatment, no significant changes in MYC expression, at both the mRNA and protein levels, were observed in K562 cells transduced with dCas9-KRAB and sgRNAs targeted to individual BENC enhancers (E1-E5) (Supplementary Fig. 9a–c).

PVT1 is a well-known lncRNA and functions as an oncogene in many types of cancers[41]. Following targeted CRISPRi, we observed a significant reduction in H3K27ac deposition at the PVT1 locus, as well as decreased expression of the PVT1 gene (Supplementary Fig. 9d). To further test whether the altered transcription of PVT1 contributed to the regulation of cell viability/proliferation, we treated cells with antisense oligonucleotides (ASOs) targeted to PVT1 to suppress PVT1 transcription without altering H3K27ac enrichment (Supplementary Fig. 9e). No significant changes in cell growth were detected before and after ASO treatment (Supplementary Fig. 9f), suggesting that the enhancer function of PVT1, but not the transcription of PVT1, is critical for regulating the growth of BETi-resistant leukemia cells.

In parallel, we applied the CRISPR activation (CRISPRa, dCas9-p300^Core) method[42] to test whether the activation of the PVT1 locus could enhance the drug resistant capability of BETi-sensitive leukemia cells (Fig. 6f). We introduced H3K27ac modifications at the PVT1 enhancer by using the CRISPRa system in three BETi-sensitive cells, including THP1, MOLM-13 and parental AF9 AML cells (Fig. 6g). Indeed, we observed a strong induction of MYC gene expression, as well as consequent protein production, in cells treated with CRISPRa targeted to the

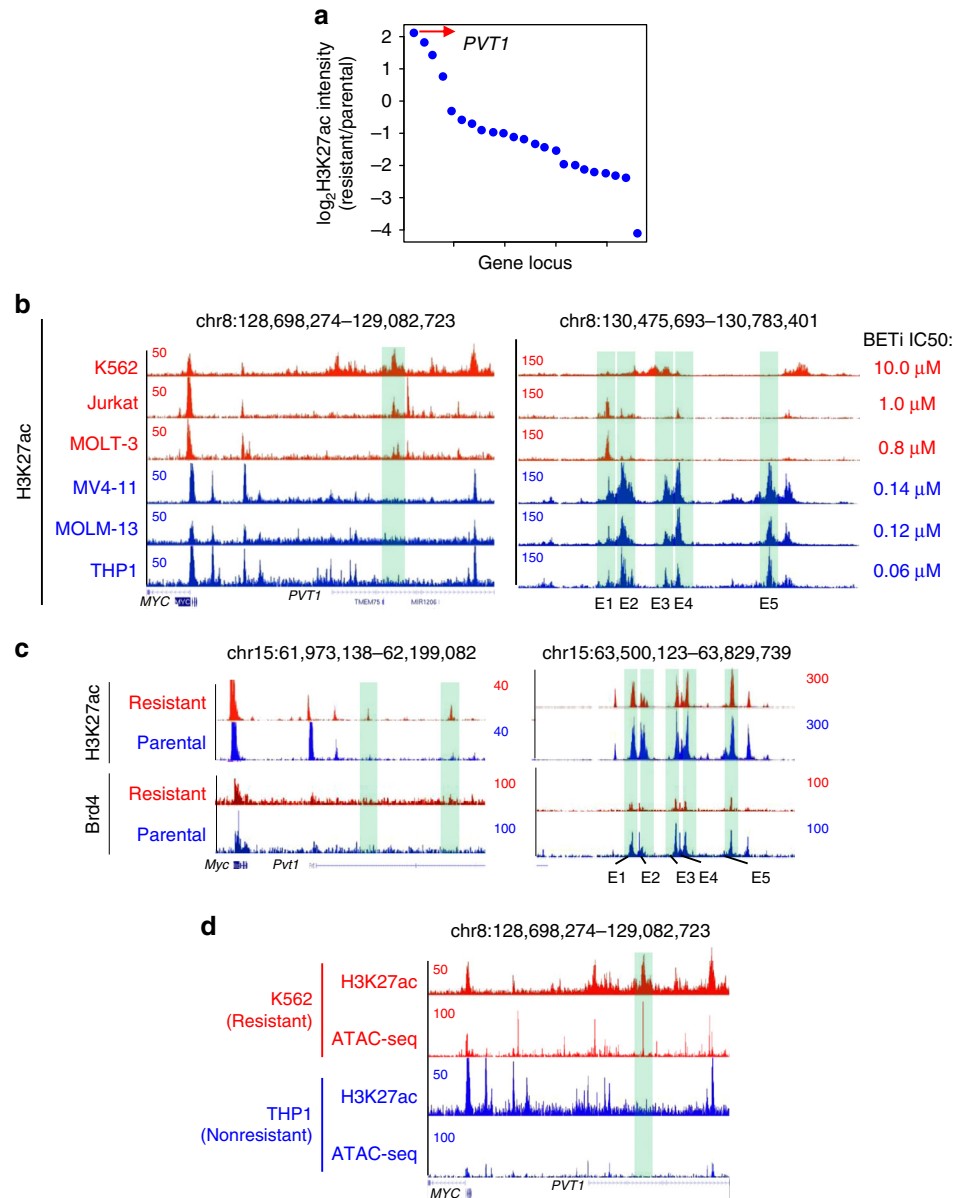

**Fig. 5 A de novo remodeled *PVT1* enhancer in BETi-resistant leukemia cells. a** Differential enrichment of H3K27ac at all identified *MYC* enhancers between parental (sensitive) and BETi-resistant murine AF9 AML cells. Red arrow: The genomic region within *Pvt1* locus at the top differential enriched regions between parental and BETi-resistant murine AF9 AML cells. **b** Genome browser views of H3K27ac enrichment in the *PVT1* locus (left) and *BENC* enhancers (E1–E5, right) of *MYC* in the indicated leukemia cell lines with relatively high (red) or low (blue) IC$_{50}$ values of BETi. Green highlighted regions represented *MYC* enhancers. **c** Genome browser view of H3K27ac enrichment at *Myc Pvt1* locus (top) and BENC enhancers (E1–E5, bottom) in BETi-sensitive (Vehicle, blue) and resistant (red) murine AF9 AML cells. Green highlighted regions were identified *MYC* enhancers. **d** Genome browser views of H3K27ac and ATAC-seq signals of *MYC PVT1* enhancers in K562 (red) and THP1 (blue) cells. Green highlighted regions were identified MYC enhancers.

*PVT1* locus (Fig. 6h, i, Supplementary Fig. 9g, h). Furthermore, compared with control groups, these high MYC-expressing cells with H3K27ac enrichment at the *PVT1* locus showed higher resistance to BETi treatment, with the IC$_{50}$ value increased by 3-fold (Fig. 6j, Supplementary Fig. 9i).

These results aligned well with the BRD4 binding data (Fig. 5c), implying that the *PVT1* enhancer, but not BENC super enhancers, is essential for re-establishing *MYC* expression in BETi-resistant leukemia cells. Collectively, results from CRISPRi and CRISPRa suggest that the *PVT1* enhancer, which is marked by the newly identified de novo H3K27ac peak in BETi-resistant leukemia cells, may experience drug-induced remodeling to

reactivate *MYC* transcription after BETi treatment, thereby contributing to BETi resistance in leukemia.

**THZ1 disrupts RNAPII loading at *MYC* enhancer-promoter loops.** To further understand how THZ1 synergized with BETi to suppress BETi-resistant leukemia, we compared the published Hi-C data collected in K562 (high IC$_{50}$ for BETi) and THP1 (low IC$_{50}$ for BETi), with a focus on analyzing the long-range chromatin interactions at the *MYC* locus. We found a BETi resistance-specific *PVT1* enhancer interacting with the *MYC* promoter in K562 cells but not in THP1 cells (Fig. 7a). We also

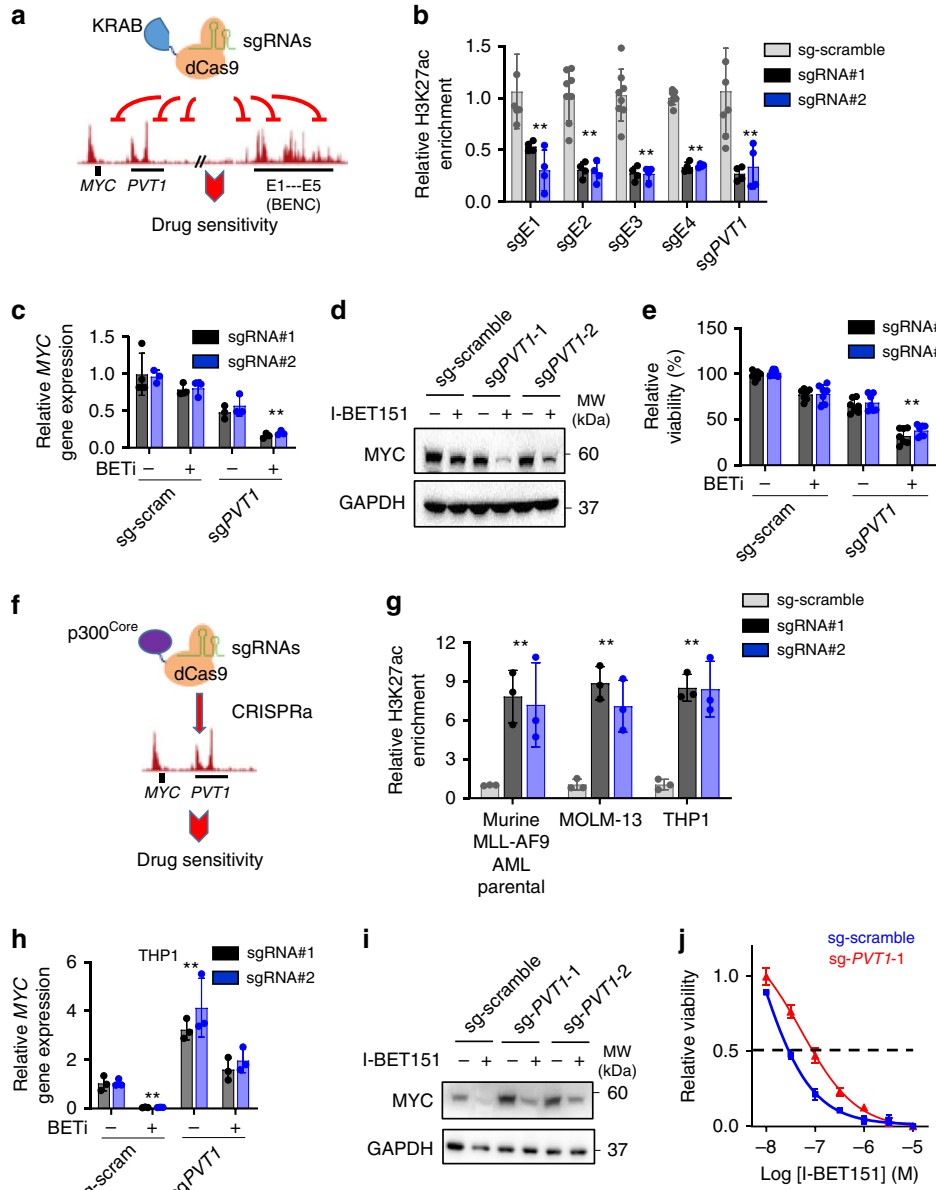

**Fig. 6 PVT1 enhancer positively regulates *MYC* expression in BETi-resistant leukemia cells. a** Schematic depicting the use of dCas9-KRAB-based CRISRPi to target MYC-associated PVT1 or BENC enhancers. **b** ChIP-qPCR analysis of H3K27ac enrichment in K562 cells transduced with a scrambled sgRNA or sgRNA targeted to the *PVT1* and *BENC* loci. Data were shown as mean ± S.D; $n = 4$ from four independent assays, **$P = 0.01$ (E1), 0.0007 (E2), 0.0006 (E3), 0.0001 (E4), and 0.01 (PVT1), by two-tailed Student's *t* test. **c–e** Transcription (**c**), protein (**d**) levels and cell viability (**e**) of K562 cells expressing dCas9-KRAB and two independent sgRNAs targeted to *PVT1* after BETi treatment. K562 cells transduced with dCas9-KRAB and scrambled sgRNAs were used as control. Data were shown as mean ± S.D; $n = 3$ from three independent assays, **$P = 0.0002$ (**c**) and 0.0004 (**e**), by two-tailed Student's *t* test. **f** Schematics depicting the use of the dCas9-p300$^{Core}$-based CRISRPa system to target MYC-associated PVT1 enhancers. **g** ChIP-qPCR analysis of H3K27ac enrichment in THP1, MOLM-13 and murine MLL-AF9 parental cells transduced with a scrambled sgRNA or sgRNA targeted to the *PVT1* locus. Data were shown as mean ± S.D; $n = 3$ from three independent assays, **$P = 0.028$ (AF9), 0.012 (MOLM-13) and 0.011 (THP1) by two-tailed Student's *t* test. **h–j** Comparison of gene transcription (**h**), protein expression (**i**) levels and dose response curves (**j**) of THP1 cells expressing dCas9-p300$^{Core}$ and two independent sgRNAs targeted to *PVT1* after BETi treatment. THP1 cells transduced with dCas9-p300$^{Core}$ and scrambled sgRNAs were used as control. Data were shown as mean ± S.D; $n = 3$ from three independent assays, **$P = 0.0001$ (sg-scram) and 0.0005 (sg-PVT1), by two-tailed Student's *t* test.

examined the published H3K27ac HiChIP data in K562 cells and discovered a strong long-range interaction between the *MYC* promoter and a BETi resistance-specific *PVT1* enhancer (Fig. 7b). In parallel, we noted a strong enrichment of RNAPII at this enhancer in K562 cells, raising the possibilities that the combination therapy may (i) alter enhancer-promoter interactions and / or (ii) block the RNAPII loading at the BETi resistance-

specific *PVT1* enhancer. To clarify these two possibilities, we performed RNAPII ChIP-qPCR and 4C experiments at the *MYC* locus in BETi-resistant K562 cells under four conditions: DMSO, I-BET151, THZ1 and the combination treatment (Fig. 7c, d). We observed that the enhancer-promoter looping remained largely unaltered under all four conditions (Fig. 7c). However, compared to the other three groups, RNAPII enrichment at TSS and

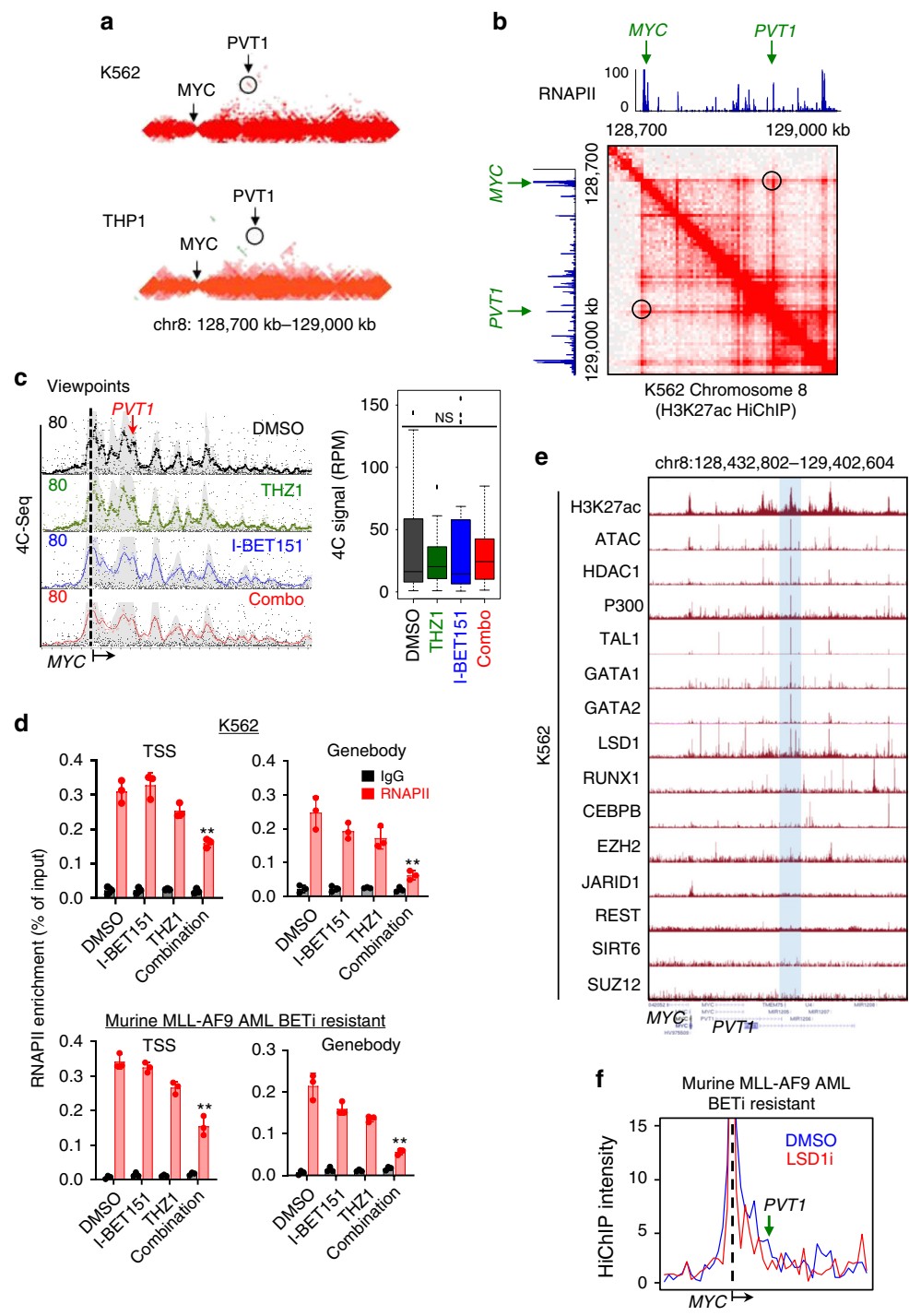

genebody was significantly reduced for the combination group in both the human K562 cells and the murine BETi-resistant AF9 AML cells (Fig. 7d). To further confirm that the transcriptional regulation activity of this enhancer remained normal, we analyzed publicly available transcriptional factors (TFs) binding data in K562 cells. We found that many important epigenetic regulators and TFs, including LSD1, GATA1, GATA2, and TAL1, were enriched at this region (Fig. 7e). Indeed, using the published HiChIP data[43], we detected significant reduction of long-range chromatin interactions between the *MYC* promoter and *PVT1* enhancers after LSD1 inhibitor (LSD1i, GSK-LSD1) treatment in murine AF9 BETi-resistant AML cells (Fig. 7f). This finding suggests that LSD1

might contribute to this long-range chromatin interaction and facilitate *MYC* transcription in BETi-resistant leukemia cells. Taken together, these data strongly suggest that the CDK7 inhibitor can block RNAPII loading at the BETi resistance-specific *PVT1* enhancer to suppress *MYC* transcription, thereby exerting a synergistic anticancer effect toward BETi-resistant leukemia.

## Discussion

Enhancer reprogramming during tumorigenesis can increase the fitness of tumor cells toward anti-tumor therapy[44,45], and thus provide an alternative strategy for tumor cells to maintain the

**Fig. 7 THZ1 inhibits *MYC* transcription by disrupting RNAPII loading at a BETi-resistant specific enhancer-promoter loop in BETi-resistant leukemia cells. a** Heatmap showing HiC signals for the K562 (top) and THP1 (bottom) cell lines at the *MYC* locus (arrow) with flanking regions (chr8: 128,700–129,000 kb). Circled region indicated the *PVT1* enhancer. **b** Genome browser views of H3K27ac HiChIP and RNAPII ChIP-seq signals within the *MYC* and *PVT1* loci. The heatmaps were generated using Juicebox. Circled regions indicated the *PTV1* enhancer. **c** 4C-seq of K562 cells treated with DMSO (black), THZ1 (green), I-BET151 (blue), and a combination of THZ1 and I-BET151 (red) for 24 h. Viewpoints were selected at the *MYC* promoter. Red arrow indicated the *PVT1* locus (left). The normalized 4C reads per fragment at the *PVT1* locus were shown as boxplots (right). $n = 3$ biological samples; NS: Not significant, by One-way ANOVA with Dunnett's post hoc correction. For the boxplots, bounds of the box spans from 25 to 75% percentile, center line within box represents median. Whiskers represent median ± 1.5 times interquartile range. Dots represent outliers. **d** ChIP-qPCR analysis of RNAPII enrichment (red) at transcription starting site (TSS) and genebody of *MYC* in K562 cells treated with DMSO, THZ1, I-BET151, and a combination of THZ1 and I-BET151 for 24 h. Chromatin pulled down with IgG was used as control (black). Data were shown as mean ± S.D; $n = 3$ from 3 independent assays. **$P = 0.014$ and 0.026 (K562, TSS and gene body); $p = 0.01$ and 0.007 (AF9, TSS and gene body), by two-tailed Student's *t* test. Drug concentrations used in Fig. 6c, d were the same as in Fig. 2b, c for each cell line. **e** Genome browser views showing the enrichment of different transcription factors (TFs) at the *PVT1* locus. Hi-C, H3K27ac HiChIP and ChIP-seq data of the corresponding TFs and RNAPII were obtained from publicly available data listed in Supplementary Table 1. **f** H3K27ac HiChIP intensities at the *PVT1* and *MYC* loci in BETi-resistant murine MLL-AF9 AML cells treated with DMSO (blue) or LSD1i (red).

expression of key genes for survival and growth during drug treatment[46]. Recent studies have shown the promise of targeting enhancer plasticity with combinatorial regimes in various BETi-resistant tumor cells, including T-ALL, triple negative breast cancer (TNBC), and neuroblastoma[23,24,47]. The rationale behinds these combinational therapies are straightforward: to squelch the newly formed enhancers with consequent suppression of oncogene transcription[23]. In the present study, we have unveiled a large group of genomic regions displaying altered H3K27ac enrichment that are independent of BRD4 binding in BETi-resistant leukemia. These newly formed H3K27ac-marked enhancers might mediate BRD4-independent oncogenic transcription to mediate BETi resistance. CDK7 has been shown to play an important role in regulating the transcriptional activity of enhancers, which motivated us to test the idea that CDK7 inhibitors (e.g., THZ1) might synergize with BETi to suppress BRD4-independent enhancers to overcome BETi resistance commonly seen in leukemia. Indeed, we observed strong synthetic lethality in BETi-resistant leukemia cells treated with the combination of BETi + THZ1 both in vitro and in vivo. THZ1 has been successfully used to suppress the growth of several cancer types, including T-ALL, TNBC, and neuroblastoma[30,32,33,48]. Compared with earlier studies using THZ1 alone as the anticancer regime, the dose of THZ1 in the combination therapy with BETi can be reduced to ~15% in Jurkat cell line[30]. These observations indicate that the newly formed BRD4-independed enhancers might serve as a vulnerable target for anticancer intervention in BETi-resistant leukemia. Since BETi are on clinical trials, few clinical samples are yet available to rigorously assess the enhancer remodeling in BETi-resistant leukemia cells. Our findings might provide valuable guidance to the future design of BETi clinical trials and establish the preclinical rationale for targeting leukemia with BETi-based combinatorial epigenetic therapy. Worthy to note, although the combination treatment with BETi and THZ1 exhibits high potency in suppressing the growth of BETi-resistant AML cells, long-term treatment (5-week) using these drugs resulted in animal death due to liver toxicity and suppression on the hematopoietic system. Therefore, further optimization of these drug candidates are required for reducing the off-tumor toxicity to enable future clinical applications.

At the molecular level, our comparative transcriptomic analysis unveiled that BETi and THZ1 could synergistically suppress the transcription of oncogenes (e.g., *MYC*) under the control of enhancers in BETi-resistant leukemia. Our integrative epigenomic analysis led to the discovery of a previously unrecognized BETi resistance-specific enhancer marked by H3K27ac within the noncoding lncRNA, *PVT1*[49]. By using dCas9-KRAB based epigenome editing method to modulate H3K27ac levels with high precision, we have established the causal relations between

this particular enhancer and *MYC* expression in BETi-treated leukemia cells. BENC has been reported as an important super enhancer clusters for *MYC* expression during leukemia development[13]. Results from our study suggest that MYC expression is not dependent on BENC during BETi resistance. Instead, MYC expression is at least partially under the control of the newly formed enhancer within the *PVT1* locus in BETi-resistant leukemia (Fig. 7). Although BETi seems to block the enhancer-promoter interaction between BENC and the *MYC* promoter, the newly formed BETi resistance-specific enhancer provides an additional interacting site for the *MYC* promoter to reactivate MYC expression in the presence of BETi treatment. Given the fact that overexpression of MYC can only partially rescue the BETi-resistant phenotype (Fig. 4h), other targets might also be involved in this process. Indeed, we observed notable changes in the transcription of E2F targets based on our RNA-seq analysis results (Fig. 4d). Although the protein level of E2F1 remains largely affected during BETi and THZ treatment, it is possible that the DNA binding properties of E2F1 might be affected when cells are adapting to this combination treatment. Further systematic studies are needed to elucidate the underlying mechanisms. Regardless, our study provides an example on how cancer cells take advantage of the alterative enhancers to maintain the transcription of key genes that are essential for tumor survival and growth in response to anticancer treatment. Targeting these remodeled enhancers, either with small molecules or with precise epigenome editing, constitutes a potential therapeutic strategy to overcome BETi resistance during anti-leukemia treatment.

Although enhancer plasticity provides an alternative strategy to facilitate oncogene transcription, the proper loading of RNAPII still remains as an indispensable step to execute mRNA transcription. In this study, we have seen unaltered enhancer-promoter interactions between the *PVT1* enhancer and the *MYC* promoter in BETi-resistant cells treated with BETi and THZ1 either alone or in combination. Suppressed transcription of *MYC* is only visualized under the combination treatment condition because of compromised enrichment of RNAPII at TSS and genebody. Based on these data and previous studies[21,30,47], we propose a tentative model to explain the drug resistance seen in BETi-treated leukemia (Fig. 8). Although BETi suppresses *MYC* expression by interrupting BRD4 binding at the BENC super enhancer in BETi-resistant leukemia cells[47], newly formed enhancer at the *PVT1* locus re-drives *MYC* expression during BETi treatment and mediate the BETi resistance. THZ1 suppresses *MYC* expression by perturbing the RNAPII activity at the newly formed PVT1 enhancer in BETi-resistant leukemia cells. Dual inhibition of BETi at BENC and THZ1 at *PVT1* enhancers blocks the interaction between the *MYC* promoter and its enhancers to suppress *MYC* transcription, thereby

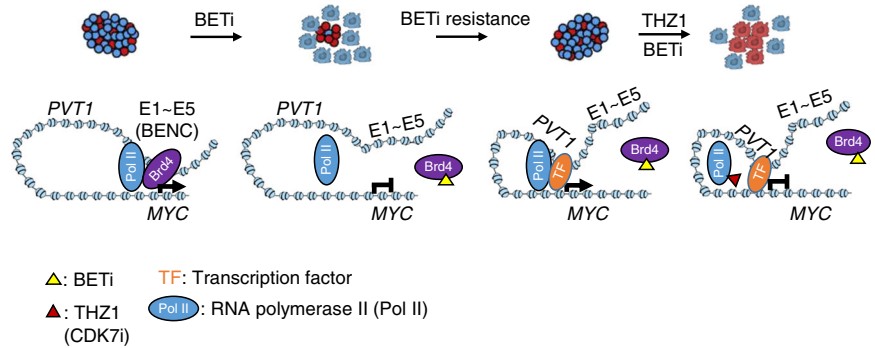

**Fig. 8 A tentative enhancer remodeling model to explain the synergistic effect of BETi and THZ1 on BETi-resistant leukemia.** *MYC* expression is regulated by the classic super-enhancer BENC (E1-E5) in BETi-sensitive leukemia cells, which is mediated by BRD4 binding. BETi blocks the BRD4 binding to its genomic targets and subsequently inhibits the expression of *MYC* and cell growth. Long-term drug treatment or primary resistance may restore *MYC* expression by enhancer remodeling: *PVT1* acts as a de novo BRD4 binding-independent enhancer, which can recruit other transcription factors and RNA Polymerase II (Pol II) to the *MYC* promoter and initiate *MYC* expression. THZ1 treatment reduces the Pol II occupancy to suppress the re-activated *MYC* transcription.

overcoming BETi resistance (Fig. 8). The mechanism unveiled in the present study does not seem to be only limited to leukemia, as we have observed many reprogrammed enhancers in other solid tumor cells during anti-tumor therapy (Supplementary Fig. 1). With the recently developed CRISPRi/CRISPRa technology, further investigations will be performed to identify cancer-associated, remodeled enhancers as potential therapeutic vulnerabilities to overcome anticancer drug resistance.

## Methods

**Cell culture and lentivirus generation.** HEK293T, K562, OCI-2 and Jurkat cell lines were purchased from ATCC and cultured in RPMI-1640 medium (Corning, Manassas, VA, USA) with 10% fetal bovine serum (Omega, Tarzana, CA, USA), 1% antibiotics (100 IU penicillin and 100 μg/mL streptomycin) and 2 mM L-glutamine. Mouse MLL-AF9 leukemia cells (BET inhibitor sensitive or resistant) were provided by Dr. Dawson's group and grown in RPMI-1640 supplemented with mouse IL-3 (10 ng/mL) and 20% FBS. Cells were routinely tested for mycoplasma contamination by PCR. For virus production, $5 × 10^6$ of HEK293T cells were plated in 10-cm plate. Lentiviral plasmids encoding targeting genes were co-transfected with pMD2.G and psPAX2 into HEK293T cells using iMFectin poly DNA transfection reagent (GenDEPOT) according to the manufacturer's instructions. Supernatant containing viral particles was collected 48 h after transfection and filtered using 0.22 μm syringe filter. For virus infection, virus and polybrene (at a final concentration of 2 μg/ml; Sigma Aldrich, Cat#107689) was added to 70% confluent cells. Fresh media were added 24 h after infection. After puromycin selection for 3 days, cells were maintained for at least one additional day without drug for further experiments.

**Immunoblotting.** Cells were lysed on ice with a RIPA lysis buffer (50 mM Tris-HCl pH 7.4, 1% NP-40, 0.5% Na- deoxycholate, 0.1% SDS, 150 mM NaCl, 2 mM EDTA, and 50 mM NaF) supplemented with protease inhibitors and phosphatase inhibitors cocktail (Gendepot, Barker, TX, USA). The concentration of the total protein was measured using the BCA protein assay kit (Thermo, Rockford, IL, USA). The protein extractions were denatured by SDS-loading buffer and incubated at 95 °C for 10 min. The denatured protein mixture was subjected to SDS-PAGE, subsequently transferred to nitrocellulose membranes (Millipore, Billerica, MA, USA), and blocked with 5% BSA in Tris-buffered saline pH 7.6 containing 0.1% Tween-20 (TBS-T). Then the membranes were probed with the corresponding primary antibodies overnight at 4 °C. After washing with TBS-T buffer for four times (5 min for each wash at room temperature), the membranes were incubated with an anti-rabbit secondary antibody (1:5,000, sigma, Cat# 7074) or an anti-mouse secondary antibody (1:3000, Cell Signaling Technology, Cat# 7076) for 2 h at room temperature, followed by washing with TBS-T for four times. The antigen–antibody complexes were detected using West-Q Pico Dura ECL Solution (Gendepot, Barker, TX, USA) and images were collected with ChemiDoc imaging system (Image Lab 6.0, BioRad, CA, USA). Detailed information regarding antibodies used in the study was listed in Supplementary Data 3. The uncropped and unprocessed scans of the blots were available in the source data file.

**Plasmids.** The shRNAs were cloned into the pLKO.1-puro vector (Addgene, #8453) by following the manufacturer's instructions. The targeting sequence of

each shRNA was listed in Supplementary Data 3. MSCV-Myc-IRES-RFP (Addgene,#35395) and pCDH-puro-cMyc (Addgene,#46970) were used to expression MYC in the rescue experiments in mouse and human cell lines, respectively. For CRISPR interference (CRISPRi) and CRISPR activation (CRIS-PRa) experiments, gRNAs were cloned into the LentiGuide-Puro (Addgene#52963) vector by using the BsmBI site. pHR-SFFV-KRAB-dCas9-P2A-mCherry (Addgene#60954) was used to express the dCas9-KRAB fusion protein. Sequences for all the sgRNAs were listed in Supplementary Data 3.

**RNA extraction and quantitative real-time PCR.** Total RNA was extracted using the Qiagen RNeasy kit and cDNA synthesis was performed with the amfiRivert Platinum One cDNA Synthesis Master Mix (Gendeport, Barker, TX, USA). Real-time quantitative PCR (qRT-PCR) was performed with a LightCycler 96 (Roche) instrument using amfiSure qGreen q-PCR Master Mix without ROX (Gendepot, Barker, TX, USA). Expression levels were determined with the delta-delta Ct method and normalized to the *GAPDH* mRNA level. Sequences of primers were provided in Supplementary Data 3.

**Drug combination assays and synergism determination.** All the chemical compounds were purchased from Selleckchem and dissolved in DMSO to make a 10 mM stock. Cells were seeded at 4000 cells per well in 96-well plates and graded concentrations of drugs with activity against different epigenetic regulators/modifiers were added to plates. Control cells were treated with DMSO. Cell viability was determined after 72 h drug treatment with a CellTiter-Glo luminescent assay (Promega). Excess over Bliss scores (score >0 indicates drug synergy, <0 indicates drug antagonism) were applied to determine synergism.

**AML mouse models and in vivo drug treatment.** Animal studies were approved by the Institutional Animal Care Use Committee (IACUC) of the Institute of Biosciences and Technology, Texas A&M University. BETi-resistant MLL-AF9-YFP murine leukemia cells ($1 × 10^6$) were injected intravenously into B6.SJL-Ptprca Pepcb/BoyJ mice (CD45.1, 6–8 weeks old; female) after sub-lethal irradiation (300 cGy). Treatment with vehicle, I-BET151, THZ1 or the combination regime (I-BET151 + THZ1) commenced after engraftment of leukemia as determined by the presence of >1% yellow fluorescent protein (YFP) in the peripheral blood of recipient mice. Mice were randomly assigned into one of the four above-mentioned treatment groups (5 mice/group). Differences in Kaplan–Meier survival curves were analyzed using the log-rank statistics. I-BET151 and THZ1 were dissolved in corn oil containing 5% (v/v) DMSO and delivered daily (5 days on, 2 days off) by intraperitoneal injection (10 mg kg$^{-1}$).

**Chromatin Immunoprecipitation (ChIP).** ChIP was performed as described previously with minor modifications[50]. Briefly, an appropriate number of cells ($1 × 10^6$ cells for H3K27Ac or $5 × 10^6$ cells for RNAPII) were harvested, crosslinked using 1% formaldehyde for 10 min and subsequently quenched with 0.125 M glycine. Cells were then lysed with lysis buffer (1% SDS, 10 mM EDTA, 50 mM Tris-HCl, pH 8.0, and protease inhibitors) and chromatin was sheared with Bioraptor to yield fragments with averaged sizes of 100–600 bp. The sonicated chromatin was centrifuged at 14,000*g* for 10 min at 4 °C. Soluble fraction of sonicated chromatin was diluted 10-fold with a ChIP-dilution buffer (16.7 mM Tris pH 7.5, 127 mM NaCl, 1.2 mM EDTA, 1% Triton X-100) and incubated with pre-mixed Dynabead of ProteinA/G with the corresponding antibodies at 4 °C overnight with gentle rotation. Chromatin-bound beads were washed with low-salt RIPA (50 mM Tris

pH 8.0, 1 mM EDTA, 150 mM NaCl, 1% Triton X-100, 0.1% SDS and 0.1% DOC), high-salt RIPA (50 mM Tris pH 8.0, 1 mM EDTA, 500 mM NaCl, 1% Triton X-100, 0.1% SDS and 0.1% DOC), LiCl wash buffer (50 mM Tris pH 8.0 250 mM LiCl, 0.5% NP-40 and 0.5% DOC) and TE (10 mM Tris pH 8.0 and 1 mM EDTA) twice each time. Beads were then resuspended in an elution buffer (10 mM Tris pH 8.0, 5 mM EDTA, 300 mM NaCl, 1% Proteinase K and 1% SDS), followed by reverse crosslinking at 65 °C for 12 h and 37 °C for 30 min with RNase A. Reverse-crosslinked DNA was purified by using a MinElute PCR purification kit. Eluted DNA was used to performed quantitative PCR (qPCR) analysis or to generate sequencing libraries using the ThruPLEX DNA-seq Kit from Rubicon Genomics (R400406) according to the manufacturer's instructions.

**CFU and CBC assay**. Primary bone marrow cells were harvested from the tibia and femur followed by red blood cell lysis buffer treatment for 1 min and centrifugation at 200g for 5 min. Pellets were suspended in PBS and used to isolate the lineage (Lin-) negative cells following the manufacturer's instructions (Biolegend, Cat#133307). Lin- cells were diluted in methylcellulose (MethoCult GF M3434; StemCell Technologies) to a concentration of $2 \times 10^4$ cells per ml. Cells were plated in a 24-well culture plate with various inhibitors or DMSO as control. CFUs were counted 10 days after seeding. For the CBC assay, peripheral venous blood was obtained from retro-orbital sinus. The analysis was performed using a hematology analyzer (ABX Pentra 60C+, HORIBA Medical) following the instructions from the manufacturer.

**Serum Alanine Transaminase (ALT) Assay**. Peripheral blood (50 μL) was collected from retro-orbital sinus of the mice. The sera were collected using a standard protocol by clotting for 30 min at room temperature followed by centrifugation at 2000 g for 15 min. The clear supernatant was collected. The serum alanine transaminase (ALT) enzymatic activity was measured using an Alanine Transaminase Colorimetric Activity Assay Kit (Cayman chemical, #700260) following the manufacturer's instructions.

**Histological analysis**. Collected tissues were fixed in 4% poly formaldehyde (PFA) for 24 h and treated with gradient alcohol dehydration. Tissue blocks were embedded in paraffin and cut into 5 μm sections for Hematoxylin and Eosin (H&E) staining. Bones were fixed in 4% formaldehyde (FA) and then decalcified for 12 h at 4 °C overnight before paraffin embedding. H&E staining was performed by following routine procedures[51,52]. Peripheral blood smear was performed on a glass slide and stained with the Wright-Giemsa reagent. Stained sections were imaged using a Nikon Eclipse Ci microscopy.

**Flow cytometry analysis**. Cells were re-suspended in FACS buffer (PBS with 1% BSA, 2 mM EDTA) and incubated with an Fc blocker for 10 min on ice. Cells were then incubated with desired antibodies at optimal concentrations for 20 min on ice in the dark followed by washing with FACS buffer twice. Then cells were re-suspended in 200 μl FACS buffer for flow cytometry analysis (LSRII, BD biosciences). Cell apoptosis rates were determined using an Annexin V kit (BD Biosciences) according to the manufacturer's instructions. For cell cycle analysis, cells were fixed by 70% ethanol/PBS overnight at −30 °C. Before flow cytometry analysis, cells were incubated at 37 °C for 30 min in a PI staining solution (0.02 mg/ml PI in PBS, supplemented with RNase A) and incubated at room temperature for 10 min.

**RNA-seq**. Total RNA was purified with the Qiagen RNeasy kit following the manufacturer's instructions. 1 μg total RNA was subjected to mRNA selection using Poly(A)Purist™ MAG Kit (Thermo Fisher Scientific). PolyA enriched mRNA was used for RNA-seq library preparation using the NEBNext® Ultra™ Directional RNA Library Prep Kit (NEB) according to the manufacturer's instructions. Agilent High Sensitivity DNA kit (Agilent Technologies) was used to monitor the quality of libraries. The libraries were then sequenced on an Illumina NextSeq 500 instrument (75 cycles, single-end). RNA-seq data were mapped to the hg19 or mm10 genome assembly using tophat-2.1.1 with default parameters. Cufflinks and cuffdiff were used to call differentially expressed genes (DEGs) (q-value < = 0.05) among the DMSO, THZ1, I-BET151, and combination groups. In-house R scripts were used to plot the scatter plot for DEGs as described in our earlier studies[51]. DEGs functional enrichment was performed using GSEA.

**4C-seq**. K562 cells ($10^7$ cells/ group) were harvested and fixed with 2% formaldehyde in PBS containing 10% FBS for 10 min at room temperature (RT) followed by quenching with 0.125 M Glycine. After washing with PBS twice, cell pellets were gently resuspended in lysis buffer (10 mM Tris-HCl pH 8, 10 mM NaCl, 0.2%Igepal) with 1x protease inhibitors (Roche, 11697498001). Cells were incubated on ice for 30 min then washed twice with lysis buffer with no protease inhibitors. Pellets were resuspended in CutSmart buffer and incubated at 37 °C in a thermomixer at 900 RPM. The NlaIII (NEB R0125L) enzyme was added at 0 h, 4 h, 16 h after incubation (200 unites per time point). 16 h after the first enzyme addition, restriction enzyme was inactivated by heating to 65 °C for 20 min. Proximity ligation was performed in a total of 1200 μL with 2000 units of T4 DNA

ligase (NEB M0202M) for 12 h at 16 °C. After ligation, samples were reversed crosslinked overnight at 68 °C. DNA was purified by the traditional phenol-chloroform extraction method followed by ethanol precipitation. Purified DNA was then digested by BfaI (NEB R0639L, 50 units) overnight followed by a second ligation using T4 DNA ligase (NEB 0202 M, 6000 units) at 16 °C overnight. DNA products were ethanol precipitated and purified by the QIAquick PCR Purification Kit (Qiagen, 28104). DNA was amplified by the Roche Expand Long Template polymerase (Roche 11759060001) with the following program: 94 °C for 2 min, 94 °C for 15 s(30 cycles); 55 °C for 1 min; 68 °C for 3 min, followed by a final step of 68 °C for 7 min. The PCR mixtures were then cleaned-up using Roche PCR purification kit (Roche 11732676001). DNA was further purified with Ampure XP beads (Agencourt A63882). Samples were then quantified with Qubit. Purified DNA was applied to generate library using the NEBNext Ultra II Kit (NEB7645) by following the manufacturer's instructions, and sequenced on an Illumina NextSeq 500 instrument. Basic4C R package was used to calculate normalized 4C signals and plot 4C signals surrounding the MYC location.

**H3K27ac ChIP-seq analysis**. H3K27ac ChIP-seq fastq files from GEO (Supplementary Data 1) or our own ChIP-seq data were downloaded to map raw reads to hg19 human genome using bowtie2[53] and the uniquely mapped reads were used for downstream analyses. MACS2[54] was used to call H3K27ac peaks. The regular enhancers were defined as the H3K27ac-enriched regions that are >5 kb away from genes and are not within any genic regions. All the regular H3K27ac enhancers were merged within six cell lines used in this study[55–59]. Normalized H3K27ac signals across merged regular enhancers were counted to generate a matrix. A matrix was applied with row as merged regular enhancers and column as cell types.

We used the ROSE pipeline (https://bitbucket.org/young_computation/rose/src/master/) for super-enhancers identification[60]. We ranked all enhancers by increasing total background-subtracted H3K27ac occupancy and plotted the total H3K27ac occupancy in the unit of total rpm/bp (reads per million per base pair). A clear point was identified in the distribution of enhancers where the occupancy signal began increasing rapidly. To define this point, the x- and y-axis were scaled from 0 to 1 and the x-axis point for which a line with a slope of 1 was tangent to the curve. We defined enhancers above this point to be super enhancers, and enhancers below as typical enhancers.

**The PCA of enhancers**. PlotPCA function was used in the DESeq2 R package[61] to perform the PCA in both regular enhancers and super enhancers. We merged all the regular H3K27ac enhancers identified from six cell lines used in this study. We then counted the normalized H3K27ac signals across these merged regular enhancers to generate a matrix. In the matrix, each row contained information regarding merged regular enhancers and each column represented the cell types. We then input this matrix to DESeq2 with default parameter to perform PCA. Similarly, the super enhancers were identified using ROSE with default setting as describe above. We merged all the super enhancers from six cell lines and performed the same PCA analysis for regular enhancers.

**Reporting summary**. Further information on research design is available in the Nature Research Reporting Summary linked to this article.

## Data availbility
The sequencing datasets have been deposited into NCBI BioProject under the accession number PRJNA543382. The source data underlying Figs. 2, 3, 4, 6, 7 and Supplementary Figs S3, S4, S6–S9 are provided as a Source Data file. A reporting summary for this Article is available as a Supplementary Information file.

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

## Acknowledgements

The authors are grateful for Dr. Jianjun Shen and the MD Anderson Cancer Center next-generation sequencing core at Smithville (CPRIT RP120348 and RP170002), and the Epigenetic core in the Institute of Biosciences and Technology at the Texas A&M University. This work was supported by grants from Cancer Prevention and Research Institute of Texas (RR140053 to Y.H., to RP170660 to Y.Z., RP180131 to D.S., RP150578 to C.S. and P.D.), National Institute of Health grants (R01HL134780 and R01HL146852 to Y.H., R01GM112003 and R01CA232017 to Y.Z.), the Welch Foundation (BE-1913-20190330 to Y.Z.), the John S. Dunn Foundation (to Y.H.), the American Cancer Society (RSG-18-043-01-LIB to YH; RSG-16-215-01-TBE to Y.Z.), Cancer Fighter of Houston to Y.H., and by an allocation from the Texas A&M University start-up funds (Y.H. and D.S.).

## Author contributions

Y.H. directed and oversaw the project. L.G. performed majority of biological experiments. H.Z. performed flow cytometry analysis. A.G. provided recipient mice and performed the sub-lethal irradiation procedures. T.L. performed histology analysis. M.L. prepared sequencing libraries. J.L. and D.S. performed integrative data analysis. M.D. provide the cell lines. C.S., P.D., M.G., Y.Z., and M.D. provided intellectual inputs and valuable guidance. Y.H., Y.Z., and G.L. wrote the manuscript. All the authors participated in the discussion, data interpretation and manuscript editing.

## Competing interests

The authors declare no competing interests.
