## [Peer Review File · Nature Communications]

Reviewers' comments:

Reviewer #1 (Remarks to the Author):

The study by Guo et al investigates strategies for overcoming acquired resistance to BET bromodomain inhibitors. In accord with prior studies, the authors first compared H3K27Ac profiles between JQ1 sensitive and insensitive cell lines and noticed distinct H3K27Ac patterns, suggesting that the enhancer landscape might play a role in determining the sensitive toward JQ1. Through a small scale screen of combining BETi with different chemical inhibitors that regulate enhancer activities, they found CDK7i in combination with BETi could confer synergistically suppressing the growth of BETi insensitive lines. This effect was then further validated in vivo by using a BETi resistant MLL-AF9 AML line. Transcriptomic analysis upon the treatment of BETi and/or CDK7i further revealed E2F and MYC targets are the main genes modulated by BETi and CDK7i synergistically. The over expression of MYC can partially overcome the growth arrest imposed by the combination drug treatment. This motivated the authors to further identify the enhancers that might restore MYC expression in BETi resistant cells, which led to the identification of PVT1 enhancer as main modulator. CRISPRi against PVT1 enhancer locus in BETi insensitive lines sensitize the cells toward BETi.

This study demonstrates that the BETi combined with CDK7i could overcome mechanisms of resistance, both in vitro and in vivo, through the suppression of MYC expression by blocking RNAPII loading at PVT1 enhancer. Overall, I found the data presented throughout this study be convincing and provides strong support for the central conclusions of this finding. The biggest weakness of this study is the prior demonstration that enhancer reprogramming promotes BETi resistance (e.g. Rathert et al, Nature 2015 and Fong et al, Nature 2015) in leukemia. Indeed, the Rathert et al study already noted that acetylation changes at the PVT1 locus were acquired in BETi resistant cells. Nevertheless, I believe this study has something new to offer in terms of discovering that targeting of CDK7 can overcome this enhancer reprogramming process.

Major Points for the authors to address:

Have the authors considered using the CRISPR-activation system to activate the PVT1 enhancer through to determine whether this would elevate the IC50 for JQ1 comparable to resistant line.

Minor Points for the authors to address:

Why does the author start with JQ1 but end with I-BET151. Wouldn't it better to start with I-BET151 at beginning (maybe I am too ignorant on this, do people test JQ1 more widely than I-BET151?).

Figure 1C/D don't have color code legend.

For Figure 1F analysis, the authors should include random gained H3K27Ac (irregardless of BRD4 binding if this is critical for the author) as a control.

The authors argued that they focus on MYC because the GSEA analysis revealed MYC targets are top candidates uniquely regulated by CDK7i in combination with BETi. But the NES score for E2F targets seem to be more impressive than MYC target. What was the main reason of ignoring E2F

targets, especially given the over expression of MYC can only partially rescue the phenotype?

Reviewer #2 (Remarks to the Author):

This is an interesting manuscript as the authors demonstrated that CDK7 inhibition exhibits synergistic anti-tumor effect on BET inhibitor-resistant leukemia cells. By utilizing multiple datasets of epigenetic information on various leukemia and solid tumors, they found genome-wide enhancer remodeling played a critical role in establishing BET inhibitor-resistant cancer cells at least in part due to maintaining MYC expression through a PVT1-intron enhancer to MYC promoter looping. They also showed that enhancer-mediated MYC expression was independent on BRD4, which has been shown to play a necessary role at so-called BENC-mediated MYC transcription. Although it is not certain how the epigenetic remodeling of MYC enhancers is initiated in those leukemic cells during the establishment of resistance to the BET inhibitors, this manuscript clearly revealed the synergistic effect on the resistant leukemia cells by the inhibition of both BET and CDK7 via targeting enhancer plasticity of MYC oncogene. Overall most of experiments were properly performed and this manuscript was well written, but the manuscript would be improved by addressing concerns as described below.

1. In Figure 1B, it is important to give more detailed methodology for how to generate PCA figures for enhancers and super-enhancers.

2. In order to apply the combination therapy in clinics for patients in the future, it is important to show more detailed information of how the combination therapy affects on normal hematopoiesis (e.g. complete blood counts) and also on liver function (e.g. liver enzyme levels in blood) in vivo in Figure S6. In line with this, in Figure 2, it is also helpful to determine whether the inhibition of both BRD4 and CDK7 impairs function of normal stem/progenitor cells (e.g. cord blood CD34+ cells) in in vitro setting due to suppressing expression levels of MYC and other potential target genes.

3. In Figure 3, the combination therapy gave a longer survival in mice than either single agent did, but eventually all mice died within 40 days in this setting. I am wondering whether or not the combination therapy-given mice died due to the expansion of leukemic cells suppressing normal hematopoiesis despite the reduced tumor proportion in the BM and in the spleen.

4. In Figure 5E-5I, sgPVT1 decreased level of H3K27ac and expression of MYC, but the sgPVT1 and BET inhibition showed moderate effect on leukemic cell viability in Figure 4I. I am wondering whether sgPVT1 dys-regulated expression level of PVT1, which may contribute to activating leukemia cell viability, since it has been reported to function as an oncogene.

5. In Figure 6E and Figure 7, the authors showed binding of multiple transcription factors and epigenetic modifiers on the enhancer of PVT1 intron in BETi-resistant K562 cells, and proposed a model of generation of enhancer remodeling for MYC region, but it is not clear so far how those

transcription factors shown in Figure 6E contribute to the initiation and maintenance of the looping between PVT1 enhancer and MYC promotor. Does knockdown or deletion of those factors prevent the enhancer remodeling at least in the PVT1-MYC region?

Response to Comments:

We thank reviewers for their constructive suggestions, which we feel are very important to further improve our manuscript. To fully address the prior concerns and comments raised by reviewers, we have performed the recommended additional experiments and analyses. We have incorporated new data (Fig.1, 6, 7, and Fig. S1, S6, S7, S9; Table S2, See the attached **Table** below) in the revised manuscript. The major changes made in the text were shown in blue.

Reviewer #1 (Remarks to the Author):

We thank the reviewer for the insightful suggestions and supportive remarks that *“Overall, I found the data presented throughout this study be convincing and provides strong support for the central conclusions of this finding.”*

1. Have the authors considered using the CRISPR-activation system to activate the PVT1 enhancer through to determine whether this would elevate the IC₅₀ for JQ1 comparable to resistant line.

Response: To address this question, we used the CRISPRa system (dCas9-p300^{Core}) in AML cells that are sensitive to BETi treatment. We introduced H3K27ac modifications at the PVT1 enhancer identified in this study in THP1, MOLM-13 and parental AF9 AML cells followed by BETi treatment. As shown in **Figure 6F-J and NEW Figure S9G-I**, the increase of H3K27ac at PVT1 enhancers led to increased MYC expression, both at the mRNA and protein levels. Furthermore, all the three CRISPRa-engineered cell lines exhibited increased IC₅₀ for BETi treatment. These data provide compelling evidence to support the conclusion that the enhancer activity of the identified PVT1 locus is important for MYC expression and BETi resistance in AML cells. The corresponding data have been updated in Figure 6, Figure S9 (Page 10, Lines 322-331).

2. Why does the author start with JQ1 but end with I-BET151. Wouldn't it better to start with I-BET151 at beginning (maybe I am too ignorant on this, do people test JQ1 more widely than I-BET151?).

Response: We thank the reviewer for pointing out this issue. As the first well-characterized BETi, JQ1 is one of most popular drug candidates used in cell-based experiments. As such, JQ1 (but not I-BET151) has a comprehensive set of IC₅₀ data that are publicly available, thereby enabling us to perform in-silicon analysis as shown in **Figure 1A**. However, JQ1 has limited applications *in vivo* due to its short half-life (~0.9 h; PMID: 25009295). By contrast, I-BET151 has a relatively longer half-life (~3 h; PMID: 21964340) and is thus more suitable for *in vivo* applications. Furthermore, the murine

AF9 BETi resistant AML cells used in this study were established through escalating doses of I-BET151 (PMID: 26367796). To ensure that the data are consistent and comparable among different groups, we performed the key molecular and biochemical experiments using I-BET151 instead of JQ1. In addition to I-BET151, we also used two additional common BET inhibitors, OTX-015 and I-BET762, to validate that the JQ1-like phenotypes are repeatable in the in vitro combination assays (**Figure S2, S3**).

3. Figure 1C/D don't have color code legend.

Response: Figure 1C/D are scatterplots smoothed by dot density, which is different from traditional heatmap. The color for these figures represents the density of dots. As the reviewer suggested, we included the color code in the revised Figures (**Figure 1C/D, Figure S1C-E**) based on the density of analyzed dots.

4. For Figure 1F analysis, the authors should include random gained H3K27Ac (irregardless of BRD4 binding if this is critical for the author) as a control.

Response: We have followed the reviewer's suggestion to use random gained H3K27Ac (regardless of BRD4 binding) as control. To do this, we selected the genomic regions with increased H3K27ac peaks regardless of BRD4 binding in BETi resistant AML cells (n = 24,461) and randomly selected ~15,000 peaks that match the number of peaks used in **Figure 1F**. We then performed GREAT analysis using these randomly selected peaks. As shown in **Figure S1B**, we observed that these random peaks are not only enriched at genes that are important for myeloid cell function, but also at genes that are essential for hematopoiesis and lymphocyte function. We have updated the related results and descriptions in **Figure S1B** and on Page 4, Lines 123-128.

5. The authors argued that they focus on MYC because the GSEA analysis revealed MYC targets are top candidates uniquely regulated by CDK7i in combination with BETi. But the NES score for E2F targets seem to be more impressive than MYC target. What was the main reason of ignoring E2F targets, especially given the over expression of MYC can only partially rescue the phenotype?

Response: We thank the reviewer for pointing out this. Although we observed strong reduction of transcripts related to MYC and E2F in AML cells treated with BETi and THZ1, we only detected the down-regulation of MYC, but not E2F, at both transcriptional and protein levels (**Figure 4G, Figure S7A**). Therefore, we decided to focus on investigating how dual inhibition of BET and CDK7

contributes to gene regulation of MYC in BETi resistant AML cells. We agree that there might be indirect regulatory mechanisms contributing to the transcriptional regulation of E2F targets in BET and CDK7 dually inhibited AML cells. We hope that the reviewer will agree with us that exploring these mechanisms will be beyond the scope of this study. We have added the related data and descriptions in **Figure S7A** and Page 8, Line 241-246.

Reviewer #2:

We thank the reviewer for the valuable advice and supportive remarks that “Overall most of experiments were properly performed and this manuscript was well written.”

1. In Figure 1B, it is important to give more detailed methodology for how to generate PCA figures for enhancers and super-enhancers.

Response: By following the reviewer’s suggestion, we have added more details in the “Methods” section to describe the identification of H3K27ac marked enhancers and super-enhancers, as well as the PCA analysis of enhancers (page 20 line 636-662). The detailed information regarding regular enhancers and super-enhancers were listed in **Figure S1A**. The PCA analysis was performed using the plotPCA function in the DESeq2 R package with default setting. We merged all the regular H3K27ac enhancers identified from six cell lines used in this study. We then counted the normalized H3K27ac signals across these merged regular enhancers to generate a matrix (as shown in the table below). We input this matrix to DESeq2 with default parameter to perform principle component analysis (PCA). Similarly, the super enhancers were identified using ROSE with default setting as described above. We merged all the super enhancers from six cell lines and perform the same PCA analysis as regular enhancer.

H3K27ac enriched regions	Cell Type1	Cell Type2	Cell Type3	Cell Type4	Cell Type5	Cell Type6
Chr2:xx-xx	1234	10	0	345	677	22
Chr3:xx-xx	19	0	4562	234	60	1890
Chr9:xx-xx	322	980	269	406	0	998
...

2. In order to apply the combination therapy in clinics for patients in the future, it is important to show more detailed information of how the combination therapy affects on normal hematopoiesis (e.g. complete blood counts) and also on liver function (e.g. liver enzyme levels in blood) in vivo in Figure S6. In line with this, in Figure 2, it is also helpful to determine whether the inhibition of both BRD4 and CDK7

impairs function of normal stem/progenitor cells (e.g. cord blood CD34+ cells) in in vitro setting due to suppressing expression levels of MYC and other potential target genes.

Response: In order to investigate whether the combination treatment of I-BET151 and THZ1 would alter normal hematopoiesis and liver function, we performed CBC (complete blood counts) and flow cytometry analysis on peripheral blood, and monitored the serum ALT levels (readout for liver function) in the mice treated with DMSO or combination therapy (**Figure S6 and Table S2**). Within 2 weeks of treatment, no overt adverse effects were observed in the hematopoietic system in these mice. In parallel, we observed a 3-fold increase in the serum ALT levels in mice after 2-week treatment of BETi and THZ1 (**Figure S6E**).

To further test the effects of BETi and THZ1 treatment in hematopoietic stem and progenitor cells (HSPCs), we performed the in vitro CFU analysis and also measured HSPCs level in mice treated with the combination therapy at 2 weeks. We observed a slight reduction in HSPC counts in both in vivo flow cytometry analysis (**Figure S6F**) and in vitro CFU analysis (**Figure S6G-H**). However, a pronounced reduction of AF9 BETi resistant AML cells following the combination treatment was observed (**Figure S6B, S6G-H**). These data suggest that the suppressive effect of the BETi and THZ1 combination are much more pronounced in BETi resistant AML cells when compared with their effects on normal HSPCs. The related description was added on Page 7, Lines 196-216.

3. In Figure 3, the combination therapy gave a longer survival in mice than either single agent did, but eventually all mice died within 40 days in this setting. I am wondering whether or not the combination therapy-given mice died due to the expansion of leukemic cells suppressing normal hematopoiesis despite the reduced tumor proportion in the BM and in the spleen.

Response: By following the reviewer's suggestion, we measured the YFP+ tumor cells and normal blood cells in moribund mice receiving the combination therapy for 5 weeks. As shown in **Figure S6B**, very little YFP+ AML cells were detected in the mice treated with BETi and THZ1. However, there was a reduction in myeloid, lymphoid, and erythroid cell populations in the peripheral blood and bone marrow after 5 weeks treatment, as reflected in CBC (complete blood counts) and flow cytometry analysis results (**Figure S6I and Table S2**); although very mild suppressive effect of combination therapy to HSPCs was observed in both in vitro (**Figure S6G-H**) and in vivo (**Figure S6F**) experiments. Furthermore, we observed significantly increased serum ALT levels in mice at this time point. These data suggest that the mice died due to the toxicity effect of long-time drug treatment of BETi and THZ1, but not because of the expansion of leukemic cells suppressing normal hematopoiesis. Therefore, further optimization of compound chemical structure and drug delivery

method are indeed highly needed to reduce the toxicity for future application. The related description was added on Page 7, Lines 196-216.

4. In Figure 5E-5I, sgPVT1 decreased level of H3K27ac and expression of MYC, but the sgPVT1 and BET inhibition showed moderate effect on leukemic cell viability in Figure 5I. I am wondering whether sgPVT1 dys-regulated expression level of PVT1, which may contribute to activating leukemia cell viability, since it has been reported to function as an oncogene.

Response: We thank the reviewer for this insightful comment. Indeed, we observed decreased transcription of PVT1 in K562 cells transduced with dCas9-KRAB and sgRNAs targeted to the PVT1 enhancer (**Figure S9D**). To further test whether the enhancer activity or the transcriptional level of PVT1 is essential for MYC expression and BETi resistance, we transfected antisense oligonucleotides (ASOs) targeted to PVT1 transcription without altering its enhancer activity in K562 cells. We observed that although cells transfected with ASO targeted to PVT1 displayed significant reduction of PVT1 expression, no overt effect on K562 cell growth was noted (**Figure S9E-F**). These results indicate that the expression of mature PVT1 lncRNA itself is dispensable for the viability of K562 cells and the second intron of the PVT1 gene primarily acted as a MYC enhancer to regulate cell growth. With regard to the moderate inhibitory effect on leukemic cell viability, we reason that other yet-to-be-discovered targets, in addition to MYC, might also be involved in BETi resistance, which will be pursued in follow-on studies. The related description was added on Page 10, Lines 313-321.

5. In Figure 6E and Figure 7, the authors showed binding of multiple transcription factors and epigenetic modifiers on the enhancer of PVT1 intron in BETi-resistant K562 cells, and proposed a model of generation of enhancer remodeling for MYC region, but it is not clear so far how those transcription factors shown in Figure 6E contribute to the initiation and maintenance of the looping between PVT1 enhancer and MYC promoter. Does knockdown or deletion of those factors prevent the enhancer remodeling at least in the PVT1-MYC region?

Response: We thank the reviewer for the insightful suggestion. To address this question, we used LSD1 as an example to test whether transcriptional factors are involved in PVT1-MYC enhancer remodeling. Taking advantage of a well-established LSD1 inhibitor GSK-LSD1 (LSD1i), we analyzed the interaction between PVT1 enhancer and MYC promoter using H3K27ac HiChIP data (PMID: 31222014) in BETi-resistant AML cells treated with and without LSD1i. As shown in **Figure 7F**, the LSD1 inhibition resulted in a reduction of the enhancer-promoter interaction at MYC and PVT1 loci in BETi resistant AML cells, suggesting that LSD1 might be involved in regulating the enhancer remodeling process.

See the Appendix Table below for changes made in the figures (both main figures and SI figures).

Figures	Revised figures	In Response to Reviewer(s)
Figure 1	Modified Figure 1C-D	R1 (comment 3)
NEW Figure 6 (Split Figure 5 into Figure 5 and Figure 6)	Added NEW Figure 6F-J	R1 (comment 1)
Figure 7	Added NEW Figure 7F	R2 (comment 5)
Supplementary Fig. S1	Added NEW Figure S1A	R2 (comment 1)
Supplementary Fig. S1	Added NEW Figure S1B	R1 (comments 4)
Supplementary Fig. S6	Added NEW Figure S6B, E-I	R2 (comments 2-3)
Supplementary Fig. S7	Added New Figure S7A	R1 (comment 5)
Supplementary Fig. S9	Added New Figure S9D-F	R2 (comment 4)
Supplementary Fig. S9	Added New Figure S9G-I	R1 (comment 1)

REVIEWERS' COMMENTS:

Reviewer #1 (Remarks to the Author):

The authors have addressed my original comments and I support publication of this study.

Reviewer #2 (Remarks to the Author):

In this revised manuscript, the authors have clearly responded to the questions and concerns raised in my original critique.

Reviewer #1 (Remarks to the Author):

The authors have addressed my original comments and I support publication of this study.

Response: We thank the reviewer for the supportive remarks.

Reviewer #2 (Remarks to the Author):

In this revised manuscript, the authors have clearly responded to the questions and concerns raised in my original critique.

Response: We thank the reviewer for the time and expertise dedicated to reviewing our manuscript.